# Deducing subnanometer cluster size and shape distributions of heterogeneous supported catalysts

Vinson Liao [1,2], Maximilian Cohen[1,2], Yifan Wang[1,2] & Dionisios G. Vlachos [1,2] ✉

Infrared (IR) spectra of adsorbate vibrational modes are sensitive to adsorbate/metal interactions, accurate, and easily obtainable in-situ or operando. While they are the gold standards for characterizing single-crystals and large nanoparticles, analogous spectra for highly dispersed heterogeneous catalysts consisting of single-atoms and ultra-small clusters are lacking. Here, we combine data-based approaches with physics-driven surrogate models to generate synthetic IR spectra from first-principles. We bypass the vast combinatorial space of clusters by determining viable, low-energy structures using machine-learned Hamiltonians, genetic algorithm optimization, and grand canonical Monte Carlo calculations. We obtain first-principles vibrations on this tractable ensemble and generate single-cluster primary spectra analogous to pure component gas-phase IR spectra. With such spectra as standards, we predict cluster size distributions from computational and experimental data, demonstrated in the case of CO adsorption on $Pd/CeO_2(111)$ catalysts, and quantify uncertainty using Bayesian Inference. We discuss extensions for characterizing complex materials towards closing the materials gap.

Actual catalytic materials are inherently heterogeneous and consist of a distribution of sites, sizes, and shapes. Supported single-atom (SA) and subnanometer cluster catalysts have been of great interest due to their reduction in cost coupled with their notable catalytic activity and selectivity in many relevant chemistries, including, but not limited to, hydrogenation, oxidation, hydroformylation, reforming, and C-C coupling reactions[1–3]. Advances in microscopy applied to single-atom catalysts[4,5] co-existing with small clusters have revealed the complexity of these materials and their dynamic nature, especially under working conditions. Characterization, i.e., elucidating the distributions and structure-dependent catalytic performances[6], is challenging due to many factors such as low metal loadings[7], poor instrumental signal-to-noise ratios (SNR), limitations of characterization techniques, the inapplicability of certain operando measurements[8], and the inherent heterogeneity of the materials. Advances in addressing these challenges is imperative to improving catalyst characterization and eventually catalyst performance[9,10].

Excitations, probed via infrared (IR) spectroscopy[11], are sensitive to interactions between adsorbates and metals, and have been extensively used to study the structure of metal oxides, supported metal particles and metal oxides, as well as single-atom catalysts[12–14]. They can accurately probe adsorbate normal vibrational modes, account for coverage effects, and can be used in-operando. Most IR-based peaks, however, are typically assigned heuristically for relatively simple spectra following the gold standard of well-defined single crystals. Inorganic complexes in the form of homogeneous catalysts have also served as molecular analogs to mononuclear metal active sites of SA catalysts to aid in peak identification[15–17]. However, IR-deduced detailed characterization of real-world catalysts is lacking[18] due to strong interactions of the highly undercoordinated metal atoms with the support[19–21], resulting in each cluster size and shape giving a different signal that is difficult to distinguish in the sampled spectra.

First-principles calculations can help with peak interpretation, but models are limited and often consider a single active site on a

---

[1]Catalysis Center for Energy Innovation, RAPID Manufacturing Institute, Delaware Energy Institute, 221 Academy St., Newark, DE 19716, USA. [2]Department of Chemical and Biomolecular Engineering, University of Delaware, 150 Academy St., Newark, DE 19716, USA. ✉e-mail: vlachos@udel.edu

well-defined crystallographic plane. The disparity between simple models and real-world working materials is reminiscent of the well-known materials gap[22,23]. Current IR quantification methodologies to bridge this gap have found limited applicability to real-world catalysts, as they have mainly been restricted to spectra obtained from large nanoparticles (NPs). A framework introduced by Lansford et al. is restricted to spectra obtained from unsupported NPs[18], and predicts the fraction of planes and adsorbate site-types, but is unable to distinguish the heterogeneity in the distributions of clusters. Kale et al. utilized site-specific extinction coefficients with peak deconvolution, interaction, and a priori assumptions about nanoparticle size and coverage to determine the catalyst active sites[24], but again is limited to NPs in the order of tens of nanometers in diameter.

Here, we develop a two-step framework to interpret and deconvolute complex IR spectra of supported single-atoms and subnanometer cluster catalysts exposed to adsorbates using first-principles spectroscopies and data-based methods. We introduce a methodology to mitigate the computational cost of isomeric combinatorial search by predicting an ensemble of low-energy $(CO)_m/Pd_n$ structures under working conditions that contributes maximally to the spectroscopic signature. We utilize first-principles density-functional theory (DFT) calculations coupled with signal processing techniques to generate realistic, single-cluster primary spectra analogous to pure component spectra in gas-phase IR spectroscopy[25,26] for this ensemble. These primary spectra serve as calibration standards. We utilize a physics-driven surrogate model to construct realistic synthetic spectra that accounts for coverage effects to benchmark spectra deconvolution. Finally, we perform spectra deconvolution of synthetic and experimental spectra within the Bayesian Inference framework to predict cluster size distributions and quantify uncertainty stemming from DFT errors and noise. We derive a criterion for matching modeled and observed spectra using the signal-to-noise ratio (SNR). We discuss the applications to characterize complex materials under working conditions to close the materials gap. We benchmark our methodology on $Pd_n/CeO_2(111)$ ($n = 1$–$20$) exposed to carbon monoxide (CO). Our framework can accurately predict cluster size and shape distributions for both synthetic and experimental spectra and is robust to overfitting spectral peaks to noise. Our results obtained directly from the deconvolution of IR spectra with little to no a priori assumptions are consistent with those made from other characterization techniques. The methodology is an important tool in catalyst characterization toward closing the materials gap.

## Results and discussion

### Modeling overview

Here, we provide an overview of our framework for determining the sizes and shapes of supported subnanometer clusters exposed to adsorbates directly from IR spectra. Our methodology is inspired by the deconvolution of gas and liquid-phase IR spectra composed of a linear combination of pure component spectra, a consequence of the Beer-Lambert Law. The linear contribution of each component is traditionally solved through a system of linear equations via least-squares fitting. Pure component calibration spectra can be easily obtained for gas and liquid phase species (from an appropriate vendor, for example) but is almost impossible to obtain for heterogenous catalysts due to the difficulty in synthesizing samples with atomic uniformity.

Our framework is composed of two major steps: (1) generation of calibration spectra from first principles (rather than experimentally) and (2) deconvolution of spectra. Given the lack of calibration standards for heterogeneous materials, our framework utilizes computational IR frequencies and intensities to generate calibration spectra. Each of these spectra, deemed primary spectra, reflects a catalyst sample composed of a single supported cluster isomer exposed to adsorbates. However, the number of cluster/adsorbate configurations even for a single size can be huge. For instance, we estimate that computing the primary spectra for every possible isomer of $Pd_{20}/CeO_2$ saturated with CO would take years. We bypass this combinatorial search by computing a low-energy ensemble of metal/adsorbate structures at working conditions for each cluster size using various machine learning and optimization techniques. This ensemble consists of low-energy structures that are thermodynamically favorable and is the subject of first principles primary spectra calculations. This step reduces the number of first principles calculations by many orders of magnitude. Experimental spectra of real materials is then deconvoluted by solving the system of linear equations associated with the Beer Lambert Law within the Bayesian inference framework to predict cluster size and shape distributions and their associated uncertainties. The Bayesian approach, rather than the commonly used frequentist approach, propagates errors and uncertainties associated with first principles computed spectra. Figure 1 shows a schematic of the overall Bayesian spectra deconvolution framework. We benchmark our framework using a model system of $Pd_n/CeO_2(111)$ ($n = 1$–$20$) exposed to saturated CO at 323 K.

### Low-energy ensemble generation

The catalyst heterogeneity is evidenced by a distribution of cluster sizes and shapes for each respective size (hereafter, also called isomers

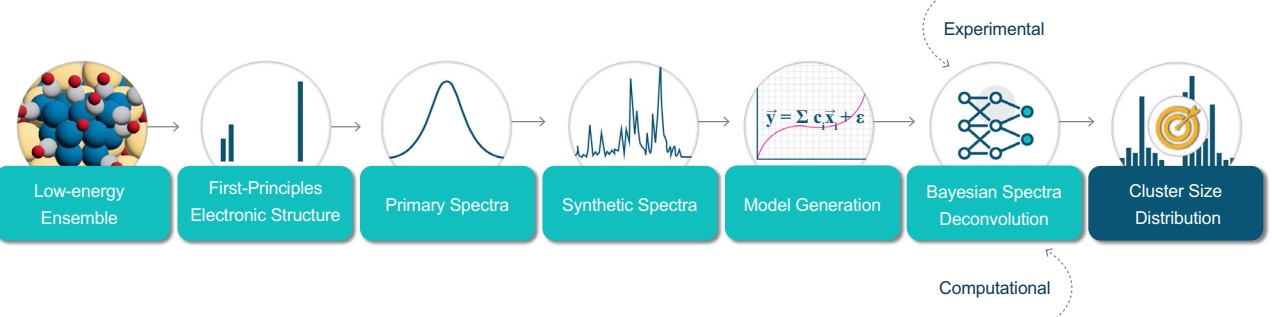

**Fig. 1 | Schematic of the Bayesian infrared spectra deconvolution procedure.** Our framework is composed of two major steps, inspired by the deconvolution of gas phase IR spectra: (1) generation of calibration spectra from first principles (rather than experimentally) and (2) deconvolution of spectra. We determine a set of low-energy structures, deemed the low-energy ensemble, of supported metal clusters exposed to adsorbates at working conditions that contribute the most to the final spectroscopic signature of the material. We compute the first-principles electronic structure to determine the IR frequencies and intensities (thus specifying the unique spectroscopic signature) for each species in the ensemble and generate primary spectra for each cluster/adsorbate configuration. Each primary spectra serves as calibration spectra for a homogenously synthesized catalyst sample. Finally, we perform deconvolution within the Bayesian Inference framework to predict the distributions of the relative fractions of each cluster size directly from experimental and computational spectra.

or structures). The number of isomers grows exponentially with size, and each isomer exposes a distribution of sites for adsorption and reaction[27]. The existence of multiple support facets and defects further enhances the heterogeneity of the material. Accounting for the combinatorics of all cluster structures and adsorbate configurations is challenging for any supported metal and adsorbate system. Determining structures directly from spectra requires solving an optimization problem to minimize the distance of computed and experimental spectra. For each trial structure generated during the optimization, adsorbate frequencies and intensities must be computed using DFT. This task is incredibly costly, and the direct structure-to-spectra matching approach is impractical. The heterogeneity of the catalyst implies that distributions rather than a single size and structure need to be accounted for, making optimization much harder. Furthermore, experimental spectrometers have limited resolution in the frequency domain, preventing the existence of an observable unique spectroscopic signature for each structure and rendering the deconvolution problem ill-posed (theoretically, with an infinite spectroscopic resolution, each potential adsorbate has a unique detectable spectroscopic signature).

To tackle these barriers, we determine the ensemble of low-energy metal/adsorbate configurations for each cluster size at a given temperature and CO partial pressure using a cluster genetic algorithm coupled with a Grand Canonical Monte Carlo (GCMC) algorithm[28]. To achieve this, one needs to develop Hamiltonians describing the metal-support, metal-metal, metal-adsorbate, and adsorbate-adsorbate (lateral) interactions using DFT and machine learning. Machine learned Hamiltonians allow for the prediction of electronic energies of arbitrary $CO-Pd/CeO_2$ structures with a minimal amount of expensive first principles calculations. The GCMC algorithm effectively minimizes the Gibbs free energy to determine the structure of the metal cluster and the distribution of surface adsorbates simultaneously at a specified temperature and CO partial pressure. This simultaneous optimization is necessary as adsorbates significantly alter the cluster structures to create preferred low-energy sites. This optimization scheme is repeated for each cluster size up to 20 Pd atoms. The low Gibbs free energy structures of each size form the low-energy ensemble that contains the most abundant structures contributing maximally to the spectral intensity.

Figure 2a shows the most energetically stable cluster/adsorbate configurations at 323 K saturated with CO for $Pd_n/CeO_2(111)$ for $n = 5–20$. We do not show Pd clusters smaller than 5 atoms as the number of possible isomers is minimal. Overall, the metal clusters have a flat or truncated pyramidal shape to maximize contact with the support especially as cluster size increases. The ratio of surface adsorbate coverage to the number of exposed surface metal atoms approaches 1:1. In addition, strong metal-support interactions also play a significant role in CO adsorption that is not captured in traditionally modeled extended surfaces. Our machine learned Hamiltonians, as well as Monte Carlo simulations, show that CO prefers to adsorb on (1) bridge and threefold sites to maximize metal coordination and (2) sites that are closer to the support for electronic stabilization. On average, our simulations show that clusters flatten under a CO environment, suggesting that the stabilization gained via the adsorption energy of CO serves as a thermodynamic driving force to offset the stability loss by overwetting of the cluster to the support.

Figure 2b shows the distributions of the Gibbs free energy normalized by the number of Pd atoms as a function of the cluster size. The free energies are referenced to a CO reservoir and calculated according to Eq. (2) of the Methods. The entropic contributions to the free energies can be decomposed into the respective configurational and vibrational contributions. We ignored vibrational entropy contributions to the free energy differences, as the change in vibrational entropy of adsorbed CO on different sites is typically less than 0.03 eV at 323 K on metals[29–32]. Configurational entropy is explicitly accounted

for by the Metropolis sampling scheme. The Gibbs free energies vary widely (from −3.0 to −1.0 eV/atom) for the same size clusters and with varying sizes due to the differences in the number of available surface sites and site-types for different isomers. Notably, the structures of the most stable Pd clusters with adsorbed CO differ from that of the bare clusters. For example, the most energetically stable isomer of bare $Pd_{20}/CeO_2$ becomes the 5th most stable isomer once CO is introduced. Literature supports the observed phenomenon; upon CO adsorption, Pd atoms diffuse and reconfigure, changing the observed structure[33–36].

To approximate the relative abundance of each cluster/adsorbate for a given size, we utilize a Boltzmann equilibrium. Figure 2c shows the ensemble probability density and Boltzmann probability density at 323 K for $Pd_{20}/CeO_2$ as a function of the normalized free energy (for $Pd_5-Pd_{19}/CeO_2$, refer to Fig. S1). Each point along the probability density curves represents a discrete minima $(CO)_m/Pd_{20}/CeO_2$ configuration sampled in the GCMC algorithm. The former refers to each discrete state being equally probable, and the latter weighted by Boltzmann statistics. The two probability density curves coincide at the limit of infinite temperature. The shaded region represents the 95% integrated probability density of the Boltzmann curve, chosen as modern FTIR spectrometers with a resolution of 2 cm$^{-1}$ typically have a signal-to-noise ratio (SNR) in the order 400 at the frequencies of the highest observed intensity peaks (i.e., C-O stretch region of 1600–2000 cm$^{-1}$). This corresponds to signal to perceived noise amplitude ratio of 20:1 (refer to Supplementary Information for more information)[37,38]. Thus, we expect 95% of the observed signal to be from the system and 5% from noise. As a result, clusters with predicted Boltzmann probabilities outside the 95% integrated probability density region contribute IR intensities indistinguishable from noise. The ensemble of structures for each cluster size within this 95% cutoff form the low-energy ensemble. For our dataset, 40 unique structures of $(CO)_m/Pd_1-Pd_{20}/CeO_2$ meet the 95% cutoff Boltzmann criterion, a remarkably small number.

We also perform an analogous Boltzmann equilibrium analysis on the bare $Pd_n/CeO_2$ clusters at an identical 323 K to determine the effect that CO has on the number of thermodynamically accessible states. Figure 3 shows the ensemble and Boltzmann probability densities for bare $Pd_{20}/CeO_2$ (for $Pd_5-Pd_{19}/CeO_2$, refer to Fig. S2). We find that the number of discrete states that meet the 95% cutoff Boltzmann criteria doubles, from 4 to 8 states, between the saturated $CO/Pd_{20}/CeO_2$ system as seen in Fig. 2c and the bare system, respectively. For the entire dataset, we find that 262 unique structures of $Pd_1-Pd_{20}/CeO_2$ meet the 95% cutoff Boltzmann criterion, almost an order-of-magnitude larger than those for $(CO)_m/Pd_1-Pd_{20}/CeO_2$. This suggests that the introduction of CO to the system leads to a thermodynamic confinement effect, limiting the number of thermodynamically accessible states at low temperatures.

## Primary spectra generation

We perform first-principles computations for the 40 configurations of $(CO)_m/Pd_n/CeO_2$ that make up the low-energy ensemble directly using DFT to construct the primary spectra. We describe the details of generating primary spectra from DFT-computed IR frequencies and intensities in the Methods section. Primary spectra are analogous to pure component spectra in gas-phase IR and are the spectroscopic signature of catalyst sample composed of a single supported cluster isomer exposed to CO. The primary spectra cannot easily be obtained experimentally due to the difficulty synthesizing homogeneous supported clusters with atomic precision. We note that DFT-computed frequencies are often systematically underestimated, and as a result, it is customary to fit linear scaling factors to experimental data to account for these errors. Linear frequency scaling factors are used for our computed primary spectra, which are optimized during the fitting procedure. Each cluster can be thought of as having a distribution of

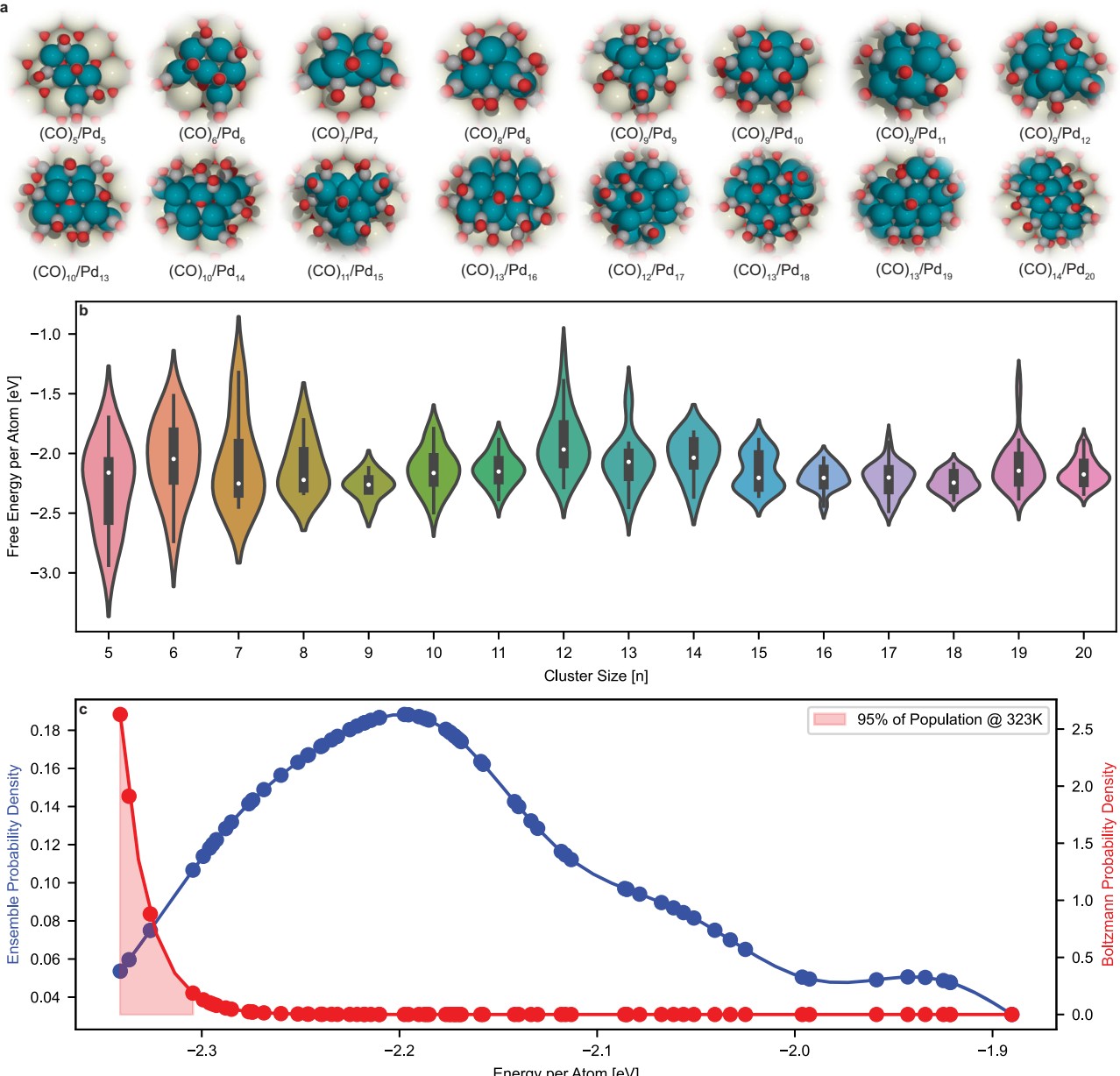

**Fig. 2 | Low energy structures versus $Pd_n/CeO_2$ cluster size for $n = 5$–20 at 323 K and saturated CO. a** Most energetically stable adsorbed structures for a given $Pd_n$/$CeO_2$. **b** Distribution of Gibbs free energies normalized by the number of Pd atoms. **c** Ensemble probability and Boltzmann probability densities of $Pd_{20}/CeO_2$ isomers saturated with CO at 323 K vs. the normalized Gibbs free energy of a configuration. Each point along the probability density curves represents a discrete $(CO)_m/Pd_{20}$/$CeO_2$ configuration. The ensemble probability density assumes each discrete state is equally probable, whereas the Boltzmann probability density weights each discrete state by its respective Boltzmann factor. The shaded red region represents the integrated 95% probability density; only 4 discrete configurations account for 95% of the isomers under working conditions. At low temperatures, relatively few discrete states are energetically accessible and dominate the ensemble of isomers compared to high temperatures.

spectroscopic signatures stemming from the uncertainty of DFT, in which the best spectra is chosen during the fitting procedure. Scaling factors computed for adsorbates on well-defined single crystals are used as informative priors to regularize and prevent overfitting. These calculated factors serve as reasonable estimates for the error in DFT frequencies. More information on the construction of linear scaling factors can be found in the Supporting Information.

Figure 4 shows primary spectra at differential CO coverage (corresponding to 1 CO per cluster) and saturated CO coverage for various Pd cluster sizes. Note that the intensities of the metal-carbon stretch region (<1000 cm⁻¹) are magnified tenfold for visibility. At differential coverages (Fig. 4a, b), it is difficult to distinguish the spectroscopic signatures of $Pd_1$ and $Pd_{10}$ as there are relatively few peaks observed.

Discerning cluster sizes at low coverages leads to high uncertainty as many combinations of single high-intensity peak spectra can form an observed IR spectra. However, at saturated CO coverage (Fig. 4c, d), multiple high-intensity peaks couple as the surface contains more adsorbates, leading to a more discernable spectroscopic signature. It is interesting that the dominant peak in the spectra of Fig. 4d (corresponding to $Pd_{20}/CeO_2$), centered in the ~1650 cm⁻¹ regime, is blue shifted when compared to the spectra in Fig. 4c (corresponding to $Pd_{10}$). This can be rationalized by CO preferentially adsorbing on lower wavenumber bridge and threefold sites on the $Pd_{20}/CeO_2$ cluster, while predominantly occupying higher wavenumber atop and bridge sites on $Pd_{10}/CeO_2$. The preferential adsorption on threefold and bridge sites on larger supported Pd clusters has also been observed in

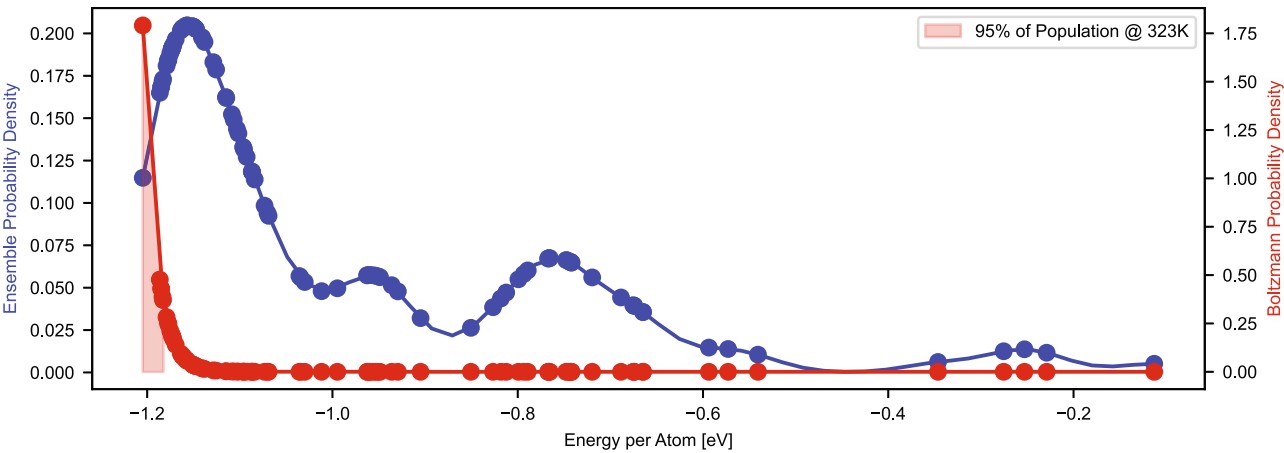

**Fig. 3 | Ensemble and Boltzmann probabilities of bare Pd₂₀/CeO₂ at 323 K.** The number of discrete states meeting the 95% cutoff criteria doubles from 4 to 8 when the system is bare versus saturated with CO. For cluster sizes between 5 and 19 atoms, we observe a range of two to ten-fold decrease in accessible states between the two systems upon exposure to CO.

the literature[28]. Thus, we choose to operate in the saturated CO coverage regime for the remainder of our work due to increase in the number of spectroscopic peaks as compared to at differential coverages.

In the Supplementary Information, we elaborate further on the effects of isomer configuration for identical sizes and CO adsorption site-types on the generated primary spectra, at both differential and saturated coverages. At differential coverages, the frequency of the highest intensity peaks (i.e., C-O stretch frequencies) is almost entirely determined by the adsorption site type (i.e., atop, bridge, hollow), as shown in Fig. S3. This trend is observed at all cluster size regimes studied, and even extends to CO frequencies at the palladium nanoparticle and single-crystal regime[39]. At saturated coverages, the spectroscopic signature of isomers of the same size exhibit large differences as the surface contains many more adsorbates than at differential coverages, and the adsorbate configurations vary greatly (Fig. S4). The ability to distinguish between different isomers further supports our decision to operate at the saturated CO coverage regime.

### Synthetic spectra generation

To benchmark our deconvolution methodology, we construct synthetic spectra representative of heterogeneous systems composed of many different cluster sizes and isomers using our primary spectra. We take advantage of the fact that IR spectral intensities obey Beers' Law and are linear with respect to the number of entities[40]. We construct synthetic spectra by taking a direct vector sum of the desired primary spectra weighted with their respective fractional contributions. Figure 5 shows an example of complex spectra of equal fractions of supported monomeric, dimeric, and trimeric Pd clusters and their individual primary cluster spectra. Intensities are normalized to ignore the effects of metal loading (and, consequently, adsorbate loading). One can see differences in the spectra with varying nuclearity; such differences allow discriminating sizes and potentially isomers. A broadening of the peaks when overlap among spectra of clusters happens is also noticeable. The applicability of this surrogate model (vs. direct DFT-computation of arbitrary heterogeneous systems) depends on the following two assumptions: (1) adsorbates on different clusters are non-interacting and (2) interacting adsorbates on the same cluster are accounted for in the primary spectra. Assumption (1) is often fulfilled for supported single atoms and clusters as metal loadings are low (i.e., high dispersion). Assumption (2) is accounted for with direct DFT computations of clusters exposed to high coverages of adsorbates.

### Spectra deconvolution via Bayesian inference (BI)

IR spectra deconvolution is traditionally difficult due to the linearly overlapping peaks of many potential candidates, each with a unique spectroscopic signature. Our Bayesian model leverages prior information of the characteristic spectral pattern and uncertainty of viable candidates for regularization to recognize overlapped signals. Expert knowledge is used to specify tighter and more informative prior distributions, which lead to narrower predicted distributions[41] (refer to Methods section and Supplementary Information for more information on the specification of prior distributions). We model the IR spectrum, $\vec{y}$, as a vector sum of wavenumber discretized primary spectra, $\vec{x}_i$, weighted by their relative fraction, $c_i$, plus some noise, $\varepsilon$:

$$\vec{y} = \sum_{i=1}^{N} c_i \vec{x}_i + \varepsilon, \varepsilon \sim N\left(0, \sigma^2 \left[\sum_{j=1}^{N} E_j \sum_{i=1}^{N} c_i \vec{x}_i e_j\right]^2\right) \quad (1)$$

The error term, $\varepsilon$, is entirely random and is intended to account for (1) background noise absent from the computational spectra, (2) DFT error in computed frequencies, and (3) spectral intensities for clusters/adsorbates not accounted for in the low-energy ensemble. We note that the DFT errors in computed frequencies, are usually systematically underestimated due to the infinite mass approximation and may not be entirely represented in the proposed mathematical form. Here, $E_j$ is the ($N$ x $N$) identity matrix (where N is the number of primary spectra considered) with 1 in position ($j,j$) and zeroes everywhere else, $e_j$ is the ($1$x $N$) row vector with 1 in position ($1,j$) and zeroes everywhere else, and σ is a scalar controlling the amount of noise in the spectra. The term $\sum_{j=1}^{N} E_j \sum_{i=1}^{N} c_i \vec{x}_i e_j$ leads to a diagonal matrix with the nonzero elements being the intensities of the reconstructed spectra, $\sum_{i=1}^{N} c_i \vec{x}_i$, at each observed frequency, without noise. This allows for a Gaussian error with standard deviation proportional (by a factor of σ) to the observed amplitude signal at each frequency to be accounted for. The scalar, σ, can assess the fit quality and is mathematically equivalent to the reciprocal of the amplitude ratio (refer to Supplementary Information for derivation). Ideally, σ should approach 0.05 as it mimics the 20:1 amplitude ratio for an observed SNR of 400 we utilize to construct our low-energy ensemble. Thus, σ allows us to infer the signal-to-noise ratio where the reconstructed spectra, $\sum_{i=1}^{N} c_i \vec{x}_i$, and

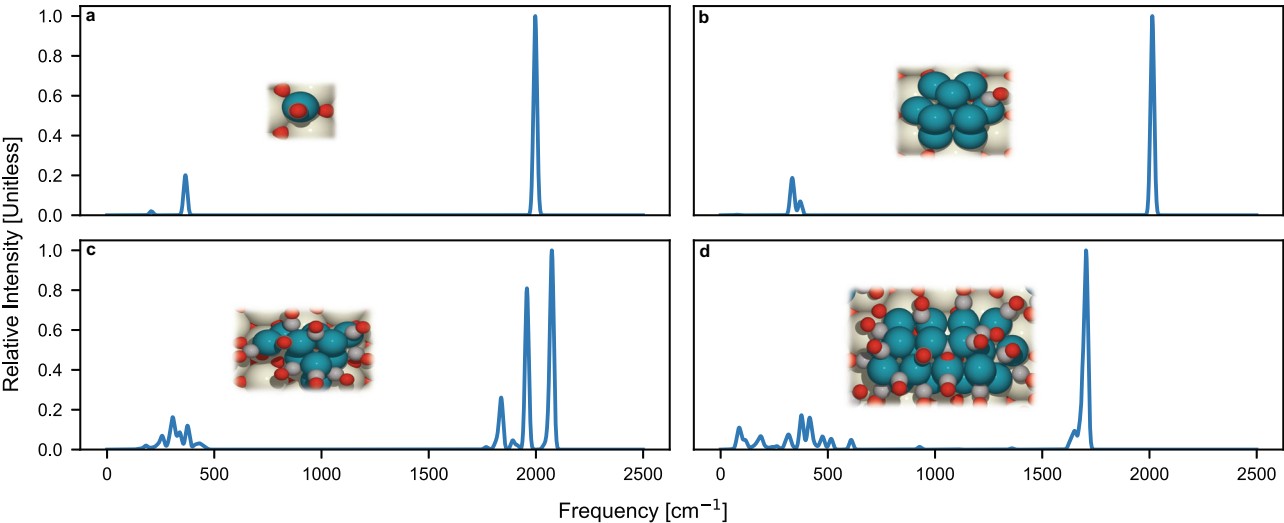

**Fig. 4 | Primary spectra of CO on various sizes of Pd/CeO₂ and CO coverages from DFT-computed frequencies and intensities.** Linear scaling factors have not been applied to these spectra. Differential coverage of CO on (**a**) Pd₁/CeO₂ and (**b**) Pd₁₀/CeO₂. Saturated coverage of CO on (**c**) Pd₁₀/CeO₂ and (**d**) Pd₂₀/CeO₂. Differential coverage refers to a single CO molecular adsorbed on the most stable adsorption site of the cluster. It is challenging to discern cluster sizes at differential coverage due to the spectra having a minimal number of unique peaks. At saturated coverage, multiple high-intensity peaks couple as the surface contains more adsorbates, leading to a discernable spectroscopic signature. The intensities of the metal-carbon stretch region (<1000 cm⁻¹) are magnified tenfold for visibility.

observed spectra, $\vec{y}$, match. Note that this equation can also be used to compare any two arbitrary spectra, $\vec{y}_1$ and $\vec{y}_2$, and their equivalent SNRs. This is useful in analyzing spectra obtained from time-resolved FTIR, for example, to determine statistically significant differences over the temporal domain. The main objective of the Bayesian Inference methodology is the estimation of the posterior distributions of each $c_i$ by iterative sampling while accounting for uncertainty in the computed primary spectra and noise (σ) in the given experimental or computational spectra. The theory and sampling methodology behind Bayesian Inference are given in Methods and Supplementary Information.

For visual simplicity, we demonstrate the deconvolution process on synthetic spectra containing equifractions of supported Pd₁, Pd₂, and Pd₃/CeO₂ saturated with CO as constructed using the surrogate model, like the one previously shown in Fig. 5. The only difference is that we introduce random Gaussian noise corresponding to an SNR of 400 (σ = 0.05) to mimic experimental spectra. Figure 6a shows the synthetic spectra, the reconstructed and deconvoluted spectra (where the means of the sampled posterior distributions are used as point estimates for the cluster fractions), and the predicted spectral noise. Note that the model does not a priori assume that Pd₄-Pd₂₀/CeO₂ is not present in the system. The intensities of the metal-carbon stretch region (<1000 cm⁻¹) are magnified tenfold for visibility.

The most stable adsorption configuration for Pd₁/CeO₂ and Pd₂/CeO₂ contain a single adsorbate on an atop site, so both primary spectra contain a single distinct peak. However, the C-O stretch frequencies are close together, and as a result, the broadened peaks overlap (Fig. 6a, blue). Without the simulated noise, a slight shoulder in the spectra can be observed to potentially distinguish the peaks (Fig. 5a), but with the conservative amount of noise introduced, heuristic assignment by the naked eye would be unable to discern them. Our framework also utilizes the information in the metal-carbon stretch region of 300–500 cm⁻¹ that is otherwise lost to further distinguish these overlapping peaks. The primary spectra of Pd₃/CeO₂ contains a doublet, with only 1 peak within the vicinity of the Pd₁ and Pd₂/CeO₂ peaks, that is easily distinguished from the other peaks. The predicted spectral noise is uncorrelated as a function of frequency and exhibits random Gaussian-like behavior, and thus suggests that the deconvolution procedure has not overfit spectral peaks to noise.

Figure 6b, c show examples of trace plots and sampled posterior distributions for the noise term, σ, and the relative concentration of Pd₁, respectively. A trace plot shows the sampled values of a particular parameter as a function of the number of iterations and is a visual way to determine how well the sampling algorithm has converged to the true posterior distribution. In general, random scatter around the median value suggests that the sampling algorithm has converged. Note that the Bayesian inference sampling methodology is inherently stochastic, so a trace plot is useful for diagnostic purposes. Also shown in the figure are the sampled posterior distributions, and the corresponding means, medians, and 95% credible intervals (CI). The mean and median of the distribution coincide and are often used as point estimates when needed. The maximum a posteriori estimation (MAP), equivalent to the distribution mode, is also often used as a point estimate but may not be appropriate for distributions that are not unimodal[42]. In this example, the mean, median, and MAP coincide and can be used as point estimates for spectra deconvolution and reconstruction. The mean value of σ = 0.054 corresponds to an equivalent SNR of 350, which is in good agreement with the specified SNR of 400 of the original synthetic spectra.

Finally, Fig. 6d shows the means and 95% CIs for each species fractions. The true values of 0.33 for Pd₁, Pd₂, and Pd₃ all lie within the 95% CIs of each distribution. The model predicts almost no clusters that are larger than Pd₃ without having evidence of this a priori. The true value of 0 is statistically difficult to sample as that value is identically the prescribed lower bound of the sampled values of $c_i$, so it does not fall within the predicted 95% CI. Our framework can estimate a distribution of the predicted metal cluster sizes on the support, but lacks detailed structural information such as local metal dispersion (i.e., heterogeneity in the distribution of the metal on the support), preferred metal adsorption sites (e.g., formation of adsorbate islands), and support defects (e.g., existence of oxygen vacancies). These local interactions that deviate from our proposed linear model are accounted for by the error term in our model and cannot be directly interpreted. Due to the limitations of our model and experimental equipment resolution, determining local spatial information directly from IR spectra is outside our current capabilities and is the scope of future work.

We also demonstrate the efficacy and robustness of our deconvolution method in the Supplementary Information over many

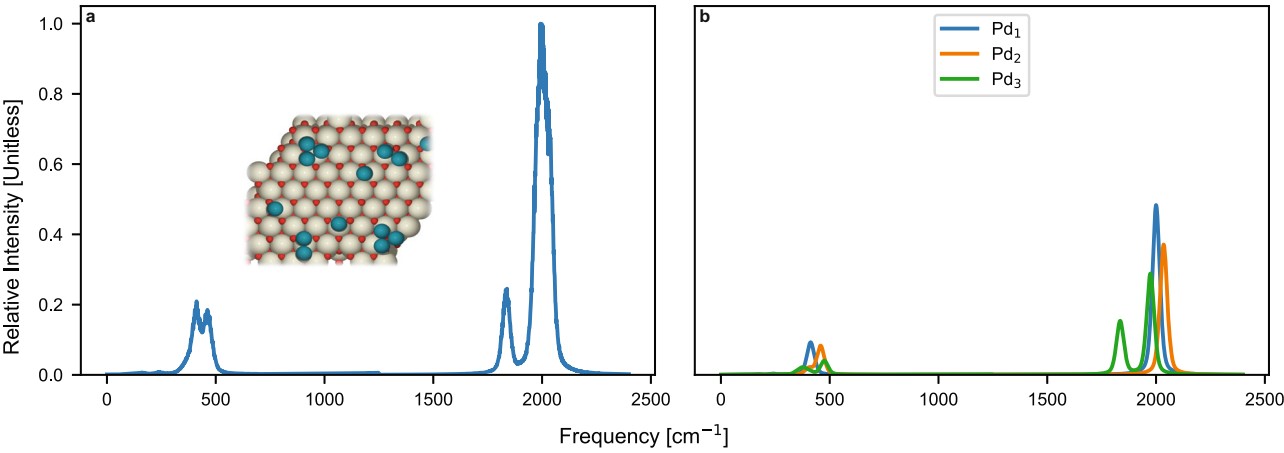

**Fig. 5 | Synthetic spectra of a system containing equal fractions of supported Pd/CeO₂ monomers, dimers, and trimers saturated with CO.** Here, the relative intensities rather than absolute intensities are shown to ignore the effects of metal loading. The intensities of the metal-carbon stretch region (<1000 cm⁻¹) are magnified tenfold for visibility. Shown are the (**a**) convoluted synthetic spectra and (**b**) original primary spectra, with intensities weighted by their relative fractions. There is a single unique isomer for each cluster size for these sizes.

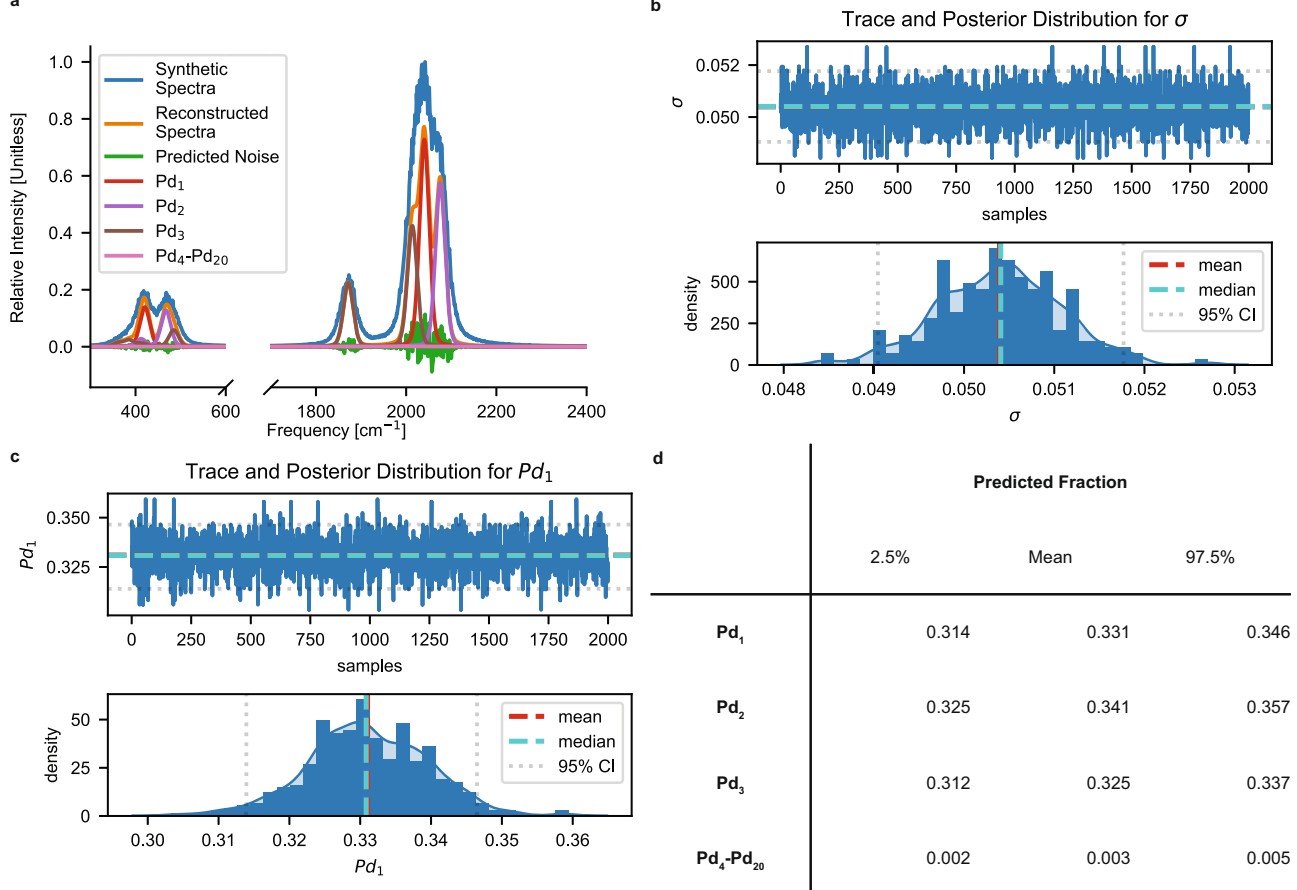

**Fig. 6 | Synthetic spectra deconvolution of a system containing equifractions of supported Pd₁, Pd₂, and Pd₃/CeO₂ saturated with CO. a** Plotted is the synthetic spectra, reconstructed and deconvoluted spectra, and the predicted spectral noise. The means of the sampled posterior distributions are used as the point estimates for the cluster fractions. The primary spectra of Pd₁ (red) and Pd₂ (purple) contain singlet peaks with severe overlap and form a single peak in the synthetic spectra that is difficult to discern by the naked eye due to noise and spectral broadening. Trace and posterior distribution plots for (**b**) σ and (**c**) fraction of Pd₁. The trace plot shows the iterations of samples drawn from the posterior. The mean value of σ = 0.054 suggests that the two spectra are equivalent for a SNR of 350, which is in good agreement with the SNR of 400 of the original synthetic spectra. **d** Means and 95% credible intervals (CI) of the predicted fractions. The model predicts almost no clusters that are larger than Pd₃.

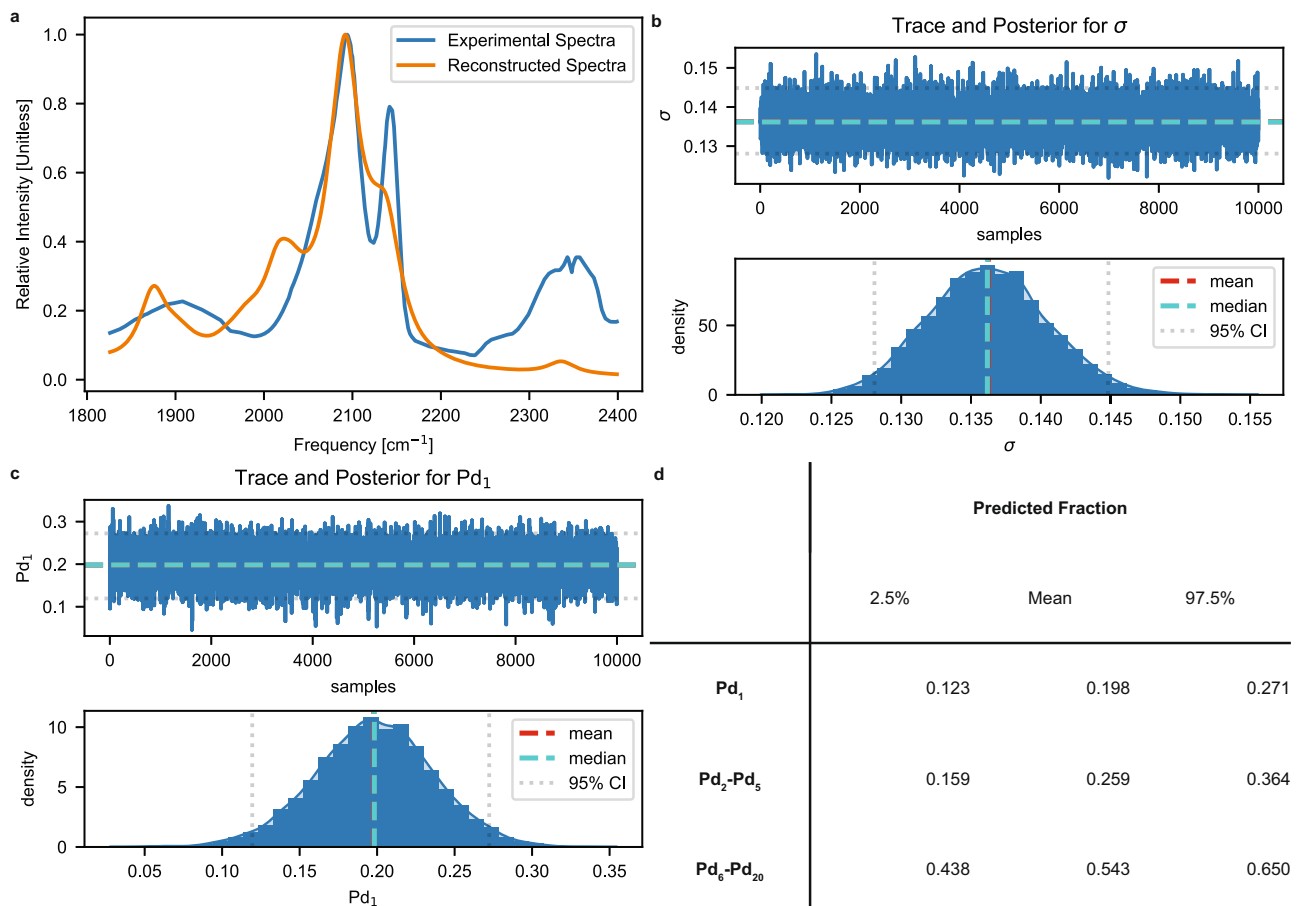

**Fig. 7 | Experimental spectra deconvolution of 1 wt% Pd/CeO$_2$ system saturated with CO at 323 K. a** Discretized experimental and reconstructed spectra. Our dataset has no spectroscopic signatures above 2200 cm$^{-1}$, which agrees with Spezzati et al.'s heuristic assignment to CO$_2$. **b** Trace and posterior distribution for σ, the error parameter. The spectra are equivalent at a SNR of 60. **c** Trace and posterior distribution plots for Pd$_1$. **d** Means and 95% credible intervals (CI) of the predicted concentrations for Pd$_1$ and bins of Pd$_2$-Pd$_5$, as well as Pd$_6$-Pd$_{20}$. We choose these bins as the former contains monolayer clusters and the latter bilayer (or larger) clusters.

synthetic spectra with randomly generated cluster fractions and varying amounts of simulated noise. Noise is simulated with signal-to-noise ratios ranging from infinity (e.g., infinitesimally small noise, the limit as σ approaches 0) to 25 (e.g., the lowest SNR of FTIR receivers reported in literature, the limit as σ approaches 0.20[37,38]) by uniformly sampling values of 0 < σ < 0.20. Note both SNR bounds are unrealistic for experimental spectra with modern day FTIR receivers and purely serve as benchmarks. A parity plot comparing MAPs of the predicted cluster fraction distribution versus true values of 100 synthetic spectra is shown in Fig. S5. We obtain a mean absolute error (MAE) of 0.049, but more importantly, the true cluster fractions lie within the 95% CI for all 100 spectra. Surprisingly, the prediction error is not correlated with σ, the amount of noise in the system, for the range of values studied. This is a good indication the model is robust enough to avoid overfitting spectra to noise.

### Experimental spectra deconvolution

Detailed experimental surface and nanocluster characterization is difficult to achieve for working materials and is often limited to simpler ordered adsorbate overlayers on single crystals[43]. We test our spectra deconvolution methodology on literature-reported IR spectra of 1 wt% Pd on CeO$_2$ nanorods saturated with CO at 323 K in which a tandem of experimental characterization techniques was used[44]. The nanorods are composed predominantly of the (111) facet of our primary spectra dataset. The published spectra provided enough detail in the C-O stretch region to be digitized, so we only utilize the frequencies and

corresponding intensities in the 1825–2400 cm$^{-1}$ range, with a discretization of 2 cm$^{-1}$. Spectroscopic information in the metal-carbon region can be helpful for overlapping peak discrimination, as shown in the previous synthetic spectra example, but is difficult to obtain in practice.

Figure 7a shows the experimental spectra and the reconstructed spectra using the means of the posterior distribution as the point estimates for the relative species concentrations. There are no spectroscopic signatures in our dataset that exceed 2200 cm$^{-1}$, so we cannot account for the broad peak centered around 2350 cm$^{-1}$. Spezzati et al. assigned this peak to CO$_2$ rather than CO/Pd/CeO$_2$, which agrees with our procedure. Our reconstructed spectra account for the major peaks at approximately 2100 and 2150 cm$^{-1}$. Figure 7b shows the trace and posterior distribution for σ, the error parameter that accounts for noise. Our reconstructed spectra have a mean σ value of 0.14 compared to the ideal value of 0.05. This suggests that the reconstructed and experimental spectra are in good agreement for an SNR of 60. Figure 7c shows the trace and posterior distribution for the Pd$_1$ fractions and suggests the presence of single atoms, with a mean of 0.198. Finally, Fig. 7d shows the means and 95% credible intervals (CI) of the predicted fractions of Pd$_1$, as well as two aggregated bins of Pd$_2$-Pd$_5$ and Pd$_6$-Pd$_{20}$. These bins were chosen to demarcate monolayer from bilayer (or larger) clusters in our dataset. Our results agree with Spezzati et al.'s TEM imaging, suggesting that Pd is highly dispersed (either as single atoms or monolayer clusters) on the support. However, we suggest that close to half of the clusters

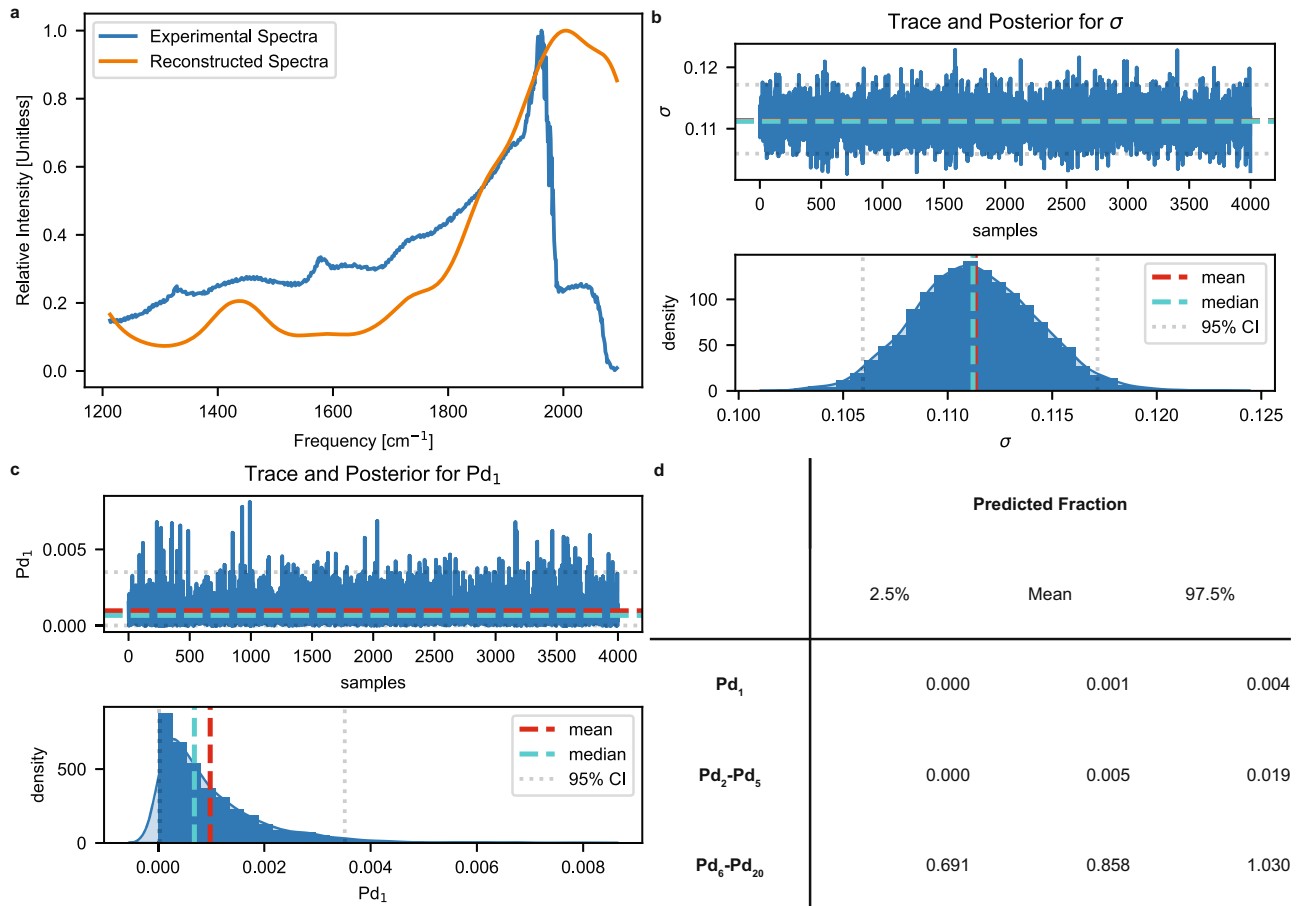

**Fig. 8 | Experimental spectra deconvolution of 5 wt% Pd/CeO$_2$ system saturated with CO at 323 K. a** Discretized experimental and reconstructed spectra. **b** Trace and posterior distribution for σ, the error parameter. The spectra are equivalent for an SNR of 80. **c** Trace and posterior distribution plots for Pd$_1$. **d** Means and 95% credible intervals (CI) of the predicted fractions for Pd$_1$ and bins of Pd$_2$-Pd$_5$, as well as Pd$_6$-Pd$_{20}$. There is little evidence to suggest single atoms or small monolayer clusters (<6 atoms) on the catalyst. We find that the supported particles are likely large multilayer particles.

may reconfigure to larger 3-dimensional clusters (Pd$_6$-Pd$_{20}$) upon exposure to CO.

We note that the oxidation state of Pd is uncertain and, as a result, the clusters may not be entirely metallic. However, there is significant evidence (by the authors and in literature) that small PdO clusters as well as single atoms can be reduced by CO at low temperatures[21], so we assume that Pd is metallic. Comparison to the experimental spectra provides further evidence for this. We also note that we model a defect free CeO$_2$, while the extent of reduction of the support of the sample in unknown due to limited characterization. The effect of oxygen vacancies on IR spectra is undoubtedly an important topic for future research.

We also benchmarked our methodology on a Pd/CeO$_2$ system with higher loadings (5 wt%) reported by Binet et al.[45]. At high loadings, we do not expect Pd to exist as single atoms or dimers/trimers due to the high probability of sintering. The catalyst is predominantly composed of (100) and (111) facets of CeO$_2$, so part of the spectra may not be accounted for in our model. We note that the sample was reduced at 423 K in H$_2$ but the authors were able to deduce, via methanol and TCNE adsorption, no observable reduction of the support. Figure 8a shows the experimental and reconstructed spectra. The reconstructed spectra account for the major peak at ~1975 cm$^{-1}$ and general spectral intensities between 1300–1900 cm$^{-1}$. Figure 8b shows the trace and posterior distribution for σ with a mean of 0.11 compared to the ideal value of 0.05. This suggests that the reconstructed and experimental spectra are in good agreement for an SNR of 80. The reconstructed spectra accounts for a large portion of the experimental spectra,

suggesting that the support may be composed mainly of CeO$_2$(111), the (111) facet may stabilize more Pd, or that the spectroscopic signatures on both facets are similar. We did not pursue this point further, but it is worth exploring in future work. The trace and posterior distribution of Pd$_1$ (Fig. 8c) show little to no evidence for single atoms. Despite the spectral intensities near 2050 cm$^{-1}$ (the calculated frequency of the C-O stretch of CO/Pd$_1$; see Fig. 4a for primary spectra) in the experimental spectra, the deconvolution process does not support the existence of single atoms. Figure 8d shows the mean and 95% credible intervals for Pd$_1$, Pd$_2$-Pd$_5$, and Pd$_6$-Pd$_{20}$. Once again, the deconvolution procedure finds little evidence for monolayer-supported clusters of less than 6 atoms. Most of the Pd atoms at high loadings exist as large multilayer nanoparticles, supported by the predicted concentrations directly from spectra.

Deducing the structure of heterogeneous single-atoms and sub-nanometer cluster catalysts has been a challenge. Surface spectroscopy, like IR, is sensitive to the sites exposed but the interpretation of experimental spectra is challenging due to the inhomogeneity of real-world world materials. The combinatorial nature of cluster shapes and sites, the DFT computational cost, and the lack of experimental methods with atomic resolution impede detailed characterization. In this work, we introduce a first principles-driven computational framework to characterize supported single-atoms and subnanometer clusters exposed to adsorbates directly from IR spectroscopic data, inspired by the deconvolution of IR spectra in the gas phase. We predict a low-energy ensemble of viable structures to reduce the combinatorial complexity of spectra deconvolution. We utilize calculations

of high-coverage adsorbate, low-energy structures to generate single-cluster primary spectra. We use state-of-the-art UHV single-crystal experiments as ground truths to correct for errors associated with DFT-computed frequencies. Finally, we perform peak deconvolution of synthetic and experimental spectra using Bayesian Inference to characterize and interpret IR spectra and derive a criterion for determining the equivalence of modeled and observed spectra using the signal-to-noise ratio. We determine cluster size distributions from computational and experimental spectra while accounting for spectral noise and uncertainties. The deconvolution procedure discriminates overlapping peaks and discerns single atoms from small clusters and large nanoparticles with results consistent with other experimental characterization techniques. Our methodology allows deduction of cluster sizes and shapes from experimental spectra without performing an unrealistic number of expensive quantum calculations. Applications in real-world materials will require an extension to many different supported facets. The general methodology presented will only improve as more accurate computational data is available.

## Methods

### Adsorbate probe molecule selection

IR spectroscopy requires the selection of an appropriate probe molecule. Carbon monoxide is extensively used due to its well-defined experimental peaks[46]. Its distinctive C-O stretch frequencies depend highly on the adsorbate site-type and local metal coordination environment and can be accurately calculated[47–49]. Carbon monoxide also does not strongly adsorb on $CeO_2(111)$; computed adsorption energies are in the order of −0.2 eV, while adsorption energies on supported Pd clusters are in the order of −2.0 eV[50]. This makes CO an ideal probe for discriminating clusters based on their corresponding spectroscopic signature.

### Low-energy ensemble generation

Enumerating and calculating the first principles vibrational frequencies and intensities of all positional combinations of cluster/adsorbates is infeasible with current computational capabilities. Rather, we determine the most energetically favorable ensemble of $(CO)_m/Pd_n/CeO_2$ configurations. We employ a machine-learned Hamiltonian to describe the energy of bare $Pd_n/CeO_2$ and a cluster genetic algorithm to predict low-energy structures[51]. We also developed a second machine-learned Hamiltonian to describe CO adsorption on $Pd_n/CeO_2$ clusters at arbitrary surface coverages that accounted for lateral interactions[28]. Both Hamiltonians were trained using DFT data. We used a rejection-free Grand Canonical Monte Carlo (GCMC) algorithm to minimize the free energy of $(CO)_m/Pd_n$ and determine the most stable adsorbate locations on low-energy clusters for given cluster size, temperature, and CO partial pressure[33]. The free energy of a specific $(CO)_m/Pd_n$ configuration referenced to a CO reservoir is given as:

$$G(T, P_{CO}, \underline{\sigma}) = E_{Pd_n/CeO_2} + E_{Pd_n/CeO_2}^{m(CO)-ads}(\underline{\sigma}) - m\left[\Delta\mu_{CO}(T, P_0) + k_B T \ln\left(\frac{P_{CO}}{P_0}\right)\right]$$

(2)

Where $E_{Pd_n/CeO_2}$ is the bare supported cluster electronic energy, $E_{Pd_n/CeO_2}^{m(CO)-ads}(\overline{\overline{\sigma}})$ is the adsorption energy of $(CO)_m$, $\mu_{CO}$ is the chemical potential of CO, and $P_0$ is a reference pressure. Zero-point energy (ZPE) corrections to the electronic energies were not needed as adsorbate frequencies were similar for identical cluster sizes, thus leading to similar ZPE corrections that cancelled out when comparing free energy differences. We ignored the vibrational contributions of the Pd atoms to reduce computational time but note that these vibrations may be important at high temperatures[52]. The GCMC algorithm (Fig. 9) thus minimizes the Gibbs free energy for a given

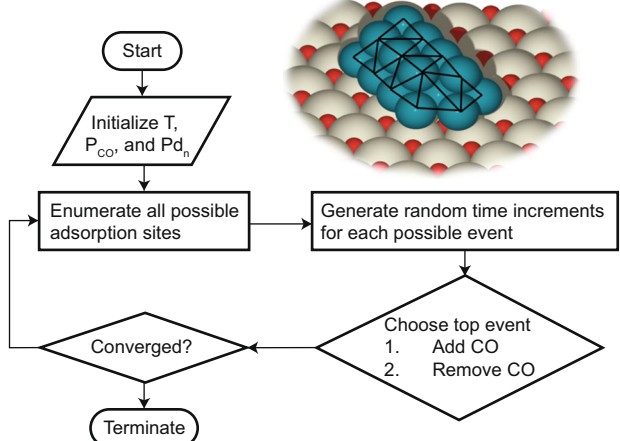

**Fig. 9 | Generation of low-energy cluster/adsorbate ensemble.** Schematic showing the steps of the rejection-free Grand Canonical Monte Carlo (GCMC) scheme, with an example of triangular mesh generated by Delaunay Triangulation to determine all surface atoms and possible adsorption sites on $Pd_{20}/CeO_2$. Vertices, edges, and centroids of the triangles correspond to atop, bridge, and hollow sites, respectively.

cluster size at a given working condition. First, the algorithm initializes a temperature, CO partial pressure, and bare $Pd_n/CeO_2$ structure. It then enumerates all possible adsorption and desorption sites. For a bare cluster, only CO may be adsorbed. We modified our previous algorithm and developed a methodology using Delaunay Triangulation to better determine all possible adsorption sites. Delaunay Triangulation generates a triangular mesh from a set of points that maximizes the enclosed volume. This triangular mesh is shown pictorially in Fig. 9. Vertices, edges, and centroids of the triangles correspond to atop, bridge, and three-fold sites, respectively. Adsorption vectors are computed as the normal vectors to the triangular faces and place adsorbates with minimal steric hindrance. We describe the Delaunay Triangulation algorithm in more detail in the Supplementary Information. The remaining GCMC algorithm remains unchanged. This modified algorithm generates realistic and optimal initial structures for DFT calculations.

### DFT calculations

Forces for the Hamiltonians and electron densities for dipole moments were obtained using Vienna ab initio Simulation Package (VASP) version 5.4 with the projector augmented wave method (PAWs)[53]. We use the PBE (Perdew-Burke-Ernzerhof) functional[54] with D3 dispersion corrections[55] as it has been used to accurately estimate frequencies for adsorbates. A Hubbard U-term was added to the PBE functional (DFT + U) employing the method by Dudarev et al.[56]. For Ce, a value of $U_{eff}$ = 4.5 eV was used as calculated by Fabris et al.[57,58]. All calculations were performed with a 400 eV plane wave cutoff and an energy convergence of $10^{-6}$ eV. For cluster calculations, a periodic $CeO_2(111)$ slab with a $(4 \times 4)$ surface unit cell of two layers thick and a vacuum gap of 15 Å was used. The bottom layer was fixed to the bulk position, and the top layer was allowed to relax. For Pd slab calculations, the model was a periodic Pd slab with $(4 \times 4)$ surface unit cell of four layers thick. Similarly, the bottom two layers were fixed to the bulk position, and the top two layers were allowed to relax. A Monkhorst-Pack $(1 \times 1 \times 1)$ and $((12/n) \times (12/m) \times 1)$ mesh were used for the Brillouin zone integration of the cluster and slabs (where n and m are the number of atoms in the x and y-directions of the slab, respectively), respectively. All input files were created using the Atomic Simulation Environment (ASE).

Frequencies corresponding to the transition from the ground to the first vibrational state were calculated using mass-weighted normal mode analysis with the harmonic approximation. VASP provides forces

for the construction of the Hessian using finite differences. A displacement of 0.015 Å for adsorbate atoms from equilibrium positions in the x, y, z directions were used for finite difference calculations of the Hessian[59]. Eigendecomposition of the Hessian provides the frequencies and directions of the vibrations from the eigenvalues and eigenvectors, respectively. The corresponding vibrational intensities are computed using the matrix product of the dipole Jacobian and normal mode eigenvectors. We employ the software CHARGEMOL, which uses the density-derived electrostatic and chemical (DDEC) approach[60–62], to integrate electronic densities from VASP to calculate the dipole moments needed.

### Primary spectra generation

We generate primary spectra from computed frequencies and intensities for a given cluster/adsorbate system as pure component spectra. Before processing the computed frequencies and intensities from DFT, it is customary to apply scaling factors to correct errors in the harmonic approximation of the potential energy surface. We use the following linear scaling factors (α) and corresponding uncertainties ($u_r$) from NIST, as shown in Eqs. (3) and (4), to adjust our frequencies.

$$\alpha = \frac{\sum_{i=1}^{n}(\nu_i{}^*\omega_i)}{\sum_{i=1}^{n}\omega_i^2} \tag{3}$$

$$u_r^2 = \frac{\sum_{i=1}^{n}(\omega_i^{2}{}^*(\alpha - \frac{\nu_i}{\omega_i}))^2}{\sum_{i=1}^{n}\omega_i^2} \tag{4}$$

where $\nu_i$ refers to experimental frequencies and $\omega_i$ refers to DFT-calculated frequencies. We utilize experimental spectra associated with experiments of well-defined adsorbate overlayer structures and known coverages on well-defined facets. Computed scaling factors, and their associated uncertainties, are used as prior distributions during the Bayesian Inference deconvolution procedure and serve as regularization for determining the best fit scaling factor for the provided experimental spectra. For more details on the computed linear scaling factors, refer to the Supplementary Information.

Mixing intensities and frequencies directly is computationally inefficient. Thus, we pre-process the scaled frequencies and intensities using a Gaussian filter to generate discretized spectra ranging from 0 to 2400 cm$^{-1}$ with a resolution of 4 cm$^{-1}$, and a peak full-width half-maximum of twice the frequency resolution to prevent significant information loss[63]. We utilize a purely Gaussian filter initially because observed random noise results in Gaussian signal response.

$$\text{Gaussian filtered spectra} = \frac{1}{\sigma\sqrt{2\pi}}\sum_{i=1}^{N}I_i e^{-\frac{(\nu_i - E)^2}{2\sigma^2}} \tag{5}$$

where σ is the standard deviation (as determined by the FWHM), $\nu_i$ and $I_i$ are the frequencies and intensities associated with a computed normal mode vibration, and E is a wavenumber vector from 0 to 2400 cm$^{-1}$, with 4 cm$^{-1}$ spacing.

We efficiently generate primary spectra of varying line shapes and line widths by convoluting the Gaussian filtered spectra with an impulse function composed of a linear combination of a Gaussian (G) and Lorentzian (L) filter, as developed by Valentine et al.[64]. This impulse function determines the final line shape and line width and depends on the full-width half-maximum (FWHM) and fraction of

Lorentzian (fL)[65].

$$G = \frac{1}{\sigma\sqrt{2\pi}}e^{-\frac{\nu^2}{2\sigma^2}}, \text{where } FWHM = 2\sigma\sqrt{2\ln(2)} \tag{6}$$

$$L = \frac{2}{\pi\sqrt{3}}(1 + \frac{4\nu^2}{3\sigma}), \text{where } FWHM = \sigma\sqrt{3} \tag{7}$$

The final impulse function is given as a linear combination of the Gaussian and Lorentzian filter, weighted by (1-fL) and fL, respectively. Finally, the impulse function is convolved using a discrete Fourier convolution with the Gaussian filtered spectra to generate the primary spectra.

### Synthetic spectra generation

To benchmark our Bayesian Inference methodology, we generate synthetic spectra by taking advantage of the fact that IR spectral intensities are linear with respect to the number of molecules. We efficiently mix spectra by applying directly summing primary spectra, each weighted by a randomly generated coefficient, $a_i$, with the uniform probability distribution:

$$P(a_i) = \begin{cases} 1, & a_i \in [0,1] \\ 0, & otherwise \end{cases} \tag{8}$$

The relative concentrations, $c_i$, for given spectra are then given by the following normalization:

$$c_i = \frac{a_i}{\sum_{i=1}^{N}a_i} \tag{9}$$

Each of these randomly generated synthetic spectra corresponds to the spectra of a sample containing relative cluster fractions given by $c_i$.

### Spectral deconvolution via Bayesian inference

Bayesian inference allows us to estimate parameters, with uncertainty, for a given dataset by providing probability distributions for each parameter of interest (in the case of our model, we estimate the probability distributions of $c_i$, FWHM, fL, α, and σ). The fundamentals of Bayesian inference are based on Bayes' Theorem, which we present in Eq. (10), for the simplest case of estimating a single parameter z given observed data x.

$$p(z|x) = \frac{p(x|z)p(z)}{p(x)} = posterior = \frac{likelihood \, x \, prior}{marginal} \tag{10}$$

Here, $p(z|x)$, known as the posterior, is the product of the likelihood, $p(x|z)$, the prior, $p(z)$, and the reciprocal of the marginal, $p(x)$. The likelihood is the probability of observing the data x given the parameter, the prior is the prior probability of parameter z, and the marginal is the probability of observing the data x. Bayesian inference allows us to estimate the posterior, $p(z,|,x)$, using Bayes' theorem for a given set of data x, and a model with parameters, z. Bayesian inference is typically computationally expensive, but there have been advances in techniques to numerically estimate the analytical form of the likelihood and prior terms[66]. We estimate the posterior distribution of the model parameters using the No-U-Turn Sampling (NUTS), an adaptive Hamiltonian Markov Chain Monte Carlo (MCMC) sampling algorithm implemented in Python with a C++ back-end for computational efficiency using the state-of-the-art Bayesian Inference package, Stan[67,68]. More details on this estimation algorithm are provided in the Supplementary Information.

## Statistical analysis

Markov Chain Monte Carlo methods, as well as other iterative sampling algorithms, converge to the target distribution at the limit of infinite simulations but rarely have strong guarantees for non-asymptotic behavior. To monitor convergence of multiple independent Markov chains, we assess the R-hat convergence diagnostic, which compares the between and within chain estimates of each sampled model parameter. This parameter assesses the how well each chain has converged to a common distribution, i.e. the true target distribution. Chains with poor inter and intra-chain agreements have R-hat values greater than 1. We only use samples with R-hat values less than 1.1, as is the recommended cutoff value reported by the original authors of the statistic in literature. For more information on the derivation of R-hat diagnostic, refer to the appropriate refs. 69,70. We utilize a minimum of 4000 total samples over a minimum of 4 Markov chains, in which half the samples are discarded as warm-up used for initialization. Once the target model parameter distributions are obtained, we utilize a two-tailed 95% credible interval for assessment, corresponding to a $p$-value of 0.05. The means of the distributions are used as point estimated for the target model parameters.

## Data availability

All data needed to evaluate the conclusions in the paper are available in the main text or the Supplementary Information. DFT data and example deconvolution code is available in the data repository on Zenodo (DOI: 10.5281/zenodo.7036103).

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

## Acknowledgements

This work was financially supported by the RAPID manufacturing institute, supported by the Department of Energy (DOE) Advanced Manufacturing Office (AMO), award number DE-EE0007888-9.5 [DGV]. RAPID projects at the University of Delaware are also made possible, in part, by funding provided by the State of Delaware. The Delaware Energy Institute acknowledges the support and partnership of the State of Delaware

in furthering the essential scientific research being conducted through the RAPID projects. This research was supported in part through the use of Information Technologies (IT) resources at the University of Delaware, specifically the high-performance computing resources.

## Author contributions

Conceptualization: V.L., D.G.V. Methodology: V.L., M.C., Y.W., D.G.V. Investigation: V.L. Visualization: V.L. Supervision: D.G.V. Writing—original draft: V.L., D.G.V. Writing—review & editing: V.L., D.G.V.

## Competing interests

The authors declare no competing interests.
