## [Peer Review File · Nature Communications]

Deducing Subnanometer Cluster Size and Shape Distributions of Heterogeneous Supported CatalystsREVIEWER COMMENTS

Reviewer #1 (Remarks to the Author):

Infrared (IR) spectroscopy is undoubtedly a very useful and sensitive tool to identify structures and adsorbate/metal interactions, relevant also for catalysis. However, it is also very difficult to obtain unbiased conclusions if there are mixtures of molecules or/and the “identity” of the molecule or cluster is not clearly defined. The authors of the present paper propose a theoretical methodology to solve this problem for small clusters as heterogeneous catalysts. In principle, this is a very timely and interesting problem. The concept, combining first principle (DFT) calculations with machine learning algorithm and statistical analysis is also a suitable methodology.

However, even the application of single-atoms or ultra-small (few atoms) as heterogeneous catalysts is questionable. However, I do not want to stress this issue further.

The individual steps of the theoretical analysis look convincing and the corresponding calculations seem to be done properly. However, the final result – see fig. 8 – the comparison between the experimental and the reconstructed spectra is disappointing! This is not surprising to me. Even the experimental spectra do not contain really meaningful information about the structure of a 5 wt% Pd/CeO₂ system saturated with CO. This is even less true for the reconstructed spectrum.

Therefore, I cannot recommend the paper for a publication in Nature Communications.

Reviewer #2 (Remarks to the Author):

The manuscript "Deducing Subnanometer Cluster Size and Shape Distributions of Heterogeneous Supported Catalysts" reports an interesting, unconventional and complex study aimed at reproducing theoretically IR spectra of supported metal nanoparticles (clusters) with adsorbed probe molecules. The main idea is to overcome the problem of describing a statistical distribution of various cluster sizes and shapes and of the corresponding IR spectra by determining low-energy structures using a combination of machine-learning, genetic algorithms, and grand canonical Monte Carlo approaches. The vibrational spectra are obtained from DFT calculations, and as a representative example the IR spectra of CO adsorbed on Pd clusters of various sizes supported on Ceria are considered as a function of CO coverage. The paper has various pros and some cons. The main positive aspect is that a conceptually novel, although rather complex, approach is proposed to generate what the authors call “synthetic” IR spectra of systems consisting of a distribution of objects to compare with corresponding experimental spectra. The amount of work performed is impressive, the method proposed is elaborate but elegant, and the comparison with experiment for the selected example is satisfactory.

The main negative aspect is that the way the paper is written makes it extremely hard to read and follow. In my view it should be entirely rewritten. In the attempt to write a concise paper the authors

have distributed essential and necessary information partly in the text, partly in the methods section at the end of the text, part in the supporting info. It does not help the reviewer the fact that also the figures are reported at the end of the paper and not as they appear in the discussion. Even the language used is not always simple and easy to follow. Once I arrived at the end of p. 6 I was so confused that I had to jump to the method section to understand what has actually been done. This is a paper that presents a new methodological approach. This is an essential part, not a complement or an appendix. The methodology used needs to be introduced in a pedagogical way, step by step, and not making reference to aspects that are important but never explained in the main text or just briefly mentioned. The paper contains also some repetition, as the text at page 3 that is largely repeated at p. 4-5. Or sentences that are hard to follow (see e.g. the sentence starting from "A trace plot shows ..." till "...indistinguishable" at p. 13). Etc.

Minor points:

The fact that CO adsorption induces restructuring of metal clusters is known since a long time, and more appropriate references exist than ref. 27

The nature of the Pd/CeO₂ interaction must be addressed in some more detail.

Looking at Figure 4a-c and Figure 4d a big shift in the CO stretching frequency is observed. This must be due to a change of adsorption site due to coverage effects. Something is said, but not clearly stated and not in the right place.

Figure 5. The spectra refer to supported clusters right? Please specify.

Figure 6. The spectra refer to gas-phase clusters? Please specify.

Page 15. The sentence "detailed experimental surface and nanocluster characterization is only achievable with ordered adsorbate overlayers on single crystals" (p. 15) is unclear or even wrong. Several examples of characterization of supported nanoclusters exist for disordered systems and are based on various techniques, not only IR.

The comparison of the simulated IR spectra with the experimental ones is done for a perfect CeO₂ surface but experimentally it is difficult to prepare defect-free ceria. It is possible that the oxygen vacancies do not affect in a major way the IR spectra, but this depends on the nature of the support, stoichiometric or reduced. What is the level of chemical reduction of the samples used to obtain the IR spectra?

In summary, this paper reports interesting and novel results, but in my view it cannot be published in present form and needs to be rewritten to large extent. I appreciate the attempt to be succinct and concise, but this must not be at the expense of clarity.

Reviewer #3 (Remarks to the Author):

Characterization of supported clusters and nanoparticles is one main challenge in heterogeneous catalysis research. The size and structure of small, supported clusters are generally unknown and difficult to assess, in particular, in the presence of reactants. IR spectroscopy is a common technique, which allows for in situ information although the interpretation of the data generally is difficult.

The manuscript describes a computational study where a technique is outlined where vibrational IR data from CO is used to obtain information on the structure of small Pd clusters supported on CeO₂. The discussed methodology is interesting and contains different steps where statistical techniques are combined with DFT calculations. The use of many structures for one cluster size is important, which considers that systems at elevated temperatures could be structurally excited. The work is suitable for Nature Communications once the Authors have addressed the following points, which mainly concern the performance of the method with respect to actual experimental data.

- 1) In the Introduction, the Authors state that IR-spectroscopy is used to study single-atom catalysts. IR-spectroscopy is a traditional technique to characterize catalysts of many types of metal particles supported on oxides; not only single-atom catalysts.
- 2) It is stated that IR-spectroscopy “occasionally” can be used for in-operando studies. IR-spectroscopy is probably one of the mostly used techniques for in-operando studies.
- 3) It is stated that assignments are done with single crystal surfaces. It is true for some cases, however, many types of well-defined system, i.e. inorganic complexes, are commonly used to aid the assignments.
- 4) It is stated that the proposed method is “an important step in intelligent catalyst design”. The method is rather a step to aid catalyst characterization.
- 5) On line 143 it is stated: “We ignore vibrational contributions to the calculated enthalpies, as the vibrational entropy of adsorbed CO is less than 0.03 eV on metals”. Perhaps the Authors are mixing enthalpy and entropy, thus it is difficult to understand what they mean. It is not clear from text if zero-point corrections are included. The adsorbed CO molecules have many modes, so the zero-point corrections should be included.
- 6) It seems as the Authors compute energies, not enthalpies.
- 7) It seems as the entropy contributions from the adsorbed CO molecules on the clusters are neglected, which is questionable.
- 8) It looks like Pd₅ can hold five CO molecules. It is written (CO)₄/Pd₅ below the figure.
- 9) On line 233 it is stated that noise is added partly to account for errors in the computed wavenumbers. The error is generally systematic (which is used in the scaling), this the DFT error is not white noise.
- 10) In Figure 7, the Authors compare their method to experiments from the literature. However, the experiment seems to have been obtained on a sample that was cooled to 50 C in the presence of O₂ (Spezzatti et al). Is Pd purely metallic when exposed to CO?
- 11) To use the computational spectral for deconvoluting the experimental spectra and obtain

information about cluster sizes it is important that the computational data can reproduce well defined systems. The Authors have used Pd(100) and Pd(111) as references and scaled wavenumbers to reproduce the experimental data. It should be mentioned in the main text that this procedure is used. Wavenumbers computed in DFT are often wrong by some 50 cm⁻¹, so a scaling procedure is needed. However, which references that have been used is not clear in the manuscript or the SI, which affects whether a proper scaling has been applied.

The computed modes are compared to experiments in the Table S1. The experimental values are taken from references 10 and 11. The original data in Ref 10 is measured in under electrochemical conditions and not used in Table S1. The data that is used is from UHV work by Bradshaw only cited in Ref. 10. Taking the example with Pd(100). Bradshaw (Surface Science 1982) states a saturation coverage of 0.56 for the peak corresponding to 1964 cm⁻¹. In the manuscript the calculations appear to have been done at a coverage of 1. For a higher coverage of 0.81, Bradshaw gives a wavenumber of 1997 cm⁻¹. This puts some doubts on the comparison to the experiments and the scaling. Because of the coupling of modes, the Authors should present the full computed spectra for the surface systems with wavenumbers and intensities.

Summary of Overall Changes in Response to Reviewer Comments

- We put our findings in the context of current literature to highlight the usefulness and importance of our developed framework for predicting catalyst structures.
- We enhance the explanations of our framework by incorporating methodological details previously found in the Methods section, as well as revamping the Modeling Overview section to provide a high-level but concise summary of our work.
- We thoroughly explain the usefulness of linear frequency scaling factors for correcting DFT computed adsorbate frequencies.
- We enhance the overall clarity of the figures, methods, and results.

Point-by-Point Comments and Responses to Reviewers' Comments

Reviewer 1

Infrared (IR) spectroscopy is undoubtedly a very useful and sensitive tool to identify structures and adsorbate/metal interactions, relevant also for catalysis. However, it is also very difficult to obtain unbiased conclusions if there are mixtures of molecules or/and the “identity” of the molecule or cluster is not clearly defined. The authors of the present paper propose a theoretical methodology to solve this problem for small clusters as heterogeneous catalysts. In principle, this is a very timely and interesting problem. The concept, combining first principle (DFT) calculations with machine learning algorithm and statistical analysis is also a suitable methodology.

Response: We appreciate the positive comments of the referee.

Comment 1

However, even the application of single-atoms or ultra-small (few atoms) as heterogeneous catalysts is questionable. However, I do not want to stress this issue further.

Response 1

Thank you for the comment. Applications of single-atoms and ultra-small clusters for heterogeneous catalysis have been extensively studied in both the literature and commercially. We have added examples in the introduction section to showcase the importance of this area of catalysis. For example, zeolites at low to moderate loading of Al and other heteroatoms are single-atom commercial catalysts but are usually not thought of as such. Small clusters are potentially present in most heterogeneous catalysts due to heterogeneity resulting from the synthesis, pre-treatment, and reactions. Thus, efficient characterization of these materials is imperative. Our change is reflected as follows:

“Supported single-atom (SA) and subnanometer cluster catalysts have been of great interest due to their reduction in cost coupled with their notable catalytic activity and selectivity in many

relevant chemistries, including, but not limited to, hydrogenation, oxidation, hydroformylation, reforming, and C-C coupling reactions¹⁻³.” (pg. 1)

Comment 2

The individual steps of the theoretical analysis look convincing and the corresponding calculations seem to be done properly. However, the final result – see fig. 8 – the comparison between the experimental and the reconstructed spectra is disappointing! This is not surprising to me. Even the experimental spectra do not contain really meaningful information about the structure of a 5 wt% Pd/CeO₂ system saturated with CO. This is even less true for the reconstructed spectrum.

Response 2

Thank you for the comment. While the experimental and reconstructed spectra may not perfectly match, we have mathematically demonstrated that these two spectra agree with each other for a realistic signal-to-noise ratio. Our methodology provides an objective, quantifiable metric for fit quality rather than a heuristic assessment. Our framework will predict and reconstruct better as experimental synthesis and characterization techniques improve. The framework can be impactful because it allows experimentalists to compare their spectra to predicted ones for various sizes and shapes to deduce possible clusters without performing quantum calculations for every size and shape imaginable. We add a brief comment in the discussion on this topic as follows:

“Our methodology allows for the comparison of experimental spectra to deduce cluster sizes and shapes without performing an unrealistic number of expensive quantum calculations.” (pg. 28)

Comment 3

Therefore, I cannot recommend the paper for a publication in Nature Communications.

Response 3

We believe we have addressed the main concern in Response 2.

Reviewer 2

The manuscript "Deducing Subnanometer Cluster Size and Shape Distributions of Heterogeneous Supported Catalysts" reports an interesting, unconventional and complex study aimed at reproducing theoretically IR spectra of supported metal nanoparticles (clusters) with adsorbed probe molecules. The main idea is to overcome the problem of describing a statistical distribution of various cluster sizes and shapes and of the corresponding IR spectra by determining low-energy structures using a combination of machine-learning, genetic algorithms, and gran canonical Monte Carlo approaches. The vibrational spectra are obtained from DFT

calculations, and as a representative example the IR spectra of CO adsorbed on Pd clusters of various sizes supported on Ceria are considered as a function of CO coverage.

The main positive aspect is that a conceptually novel, although rather complex, approach is proposed to generate what the authors call “synthetic” IR spectra of systems consisting of a distribution of objects to compare with corresponding experimental spectra. The amount of work performed is impressive, the method proposed is elaborate but elegant, and the comparison with experiment for the selected example is satisfactory.

Response: We appreciate the appraisal of our paper.

The main negative aspect is that the way the paper is written makes it extremely hard to read and follow. In my view it should be entirely rewritten. In the attempt to write a concise paper the authors have distributed essential and necessary information partly in the text, partly in the methods section at the end of the text, part in the supporting info. It does not help the reviewer the fact that also the figures are reported at the end of the paper and not as they appear in the discussion. Even the language used is not always simple and easy to follow.

Once I arrived at the end of p. 6 I was so confused that I had to jump to the method section to understand what has actually been done. This is a paper that presents a new methodological approach. This is an essential part, not a complement or an appendix. The methodology used needs to be introduced in a pedagogical way, step by step, and not making reference to aspects that are important but never explained in the main text or just briefly mentioned. The paper contains also some repetition, as the text at page 3 that is largely repeated at p. 4-5. Or sentences that are hard to follow (see e.g. the sentence starting from “A trace plot shows ...” till “...indistinguishable” at p. 13). Etc

Response: We appreciate the constructive criticism of the referee. To improve the readability of our paper, we have incorporated methodological details, previously found in the Methods, in sequential order in the context of our results. We have also significantly revamped the Modeling Overview section (pg. 4-6) to provide a high-level but concise summary of our framework. The language has been appropriately changed to make reading easier. The figures have also been reported in the text as needed rather than at the end of the paper.

Comment 1

The fact that CO adsorption induces restructuring of metal clusters is known since a long time, and more appropriate references exist than ref. 27

Response 1

Thank you for pointing this out. Additional references have been added that support CO adsorption induced restructuring of metal clusters. (pg. 9)

Comment 2

The nature of the Pd/CeO₂ interaction must be addressed in some more detail.

Response 2

Thank you for pointing this out. Our machine-learned Hamiltonians and Monte Carlo simulations give interesting insights into the strong metal-support interaction, as well as its significant role in adsorption. We find that the most important factors in determining the stability of a particular adsorption site are (1) the adsorption site type (atop, bridge, threefold) and (2) the distance of the adsorbate to the support. In particular, bridge and threefold sites are more stable than atop sites, and CO preferentially adsorbs to metal atoms closer to the support. This leads to opposing thermodynamic driving forces where, under a CO atmosphere, the cluster wants to both transform into a (1) more 3D structure where there are more bridge and threefold sites and fewer atop sites available and into a (2) flatter structure for the adsorbate to gain electronic stabilization through the support. We observe that our clusters tend to, on average, flatten under a CO environment, suggesting that the latter driving force plays a more significant role in predicting stability. This flattening is supported by experimental literature which we have cited in this same section. We comment on this observed opposing thermodynamic driving forces by adding the following statement:

“In addition, strong metal-support interactions also play a significant role in CO adsorption that is not captured in traditionally modeled extended surfaces. Our machine learned Hamiltonians, as well as Monte Carlo simulations, show that CO prefers to adsorb on (1) bridge and threefold sites to maximize metal coordination and (2) sites that are closer to the support for electronic stabilization. On average, our simulations show that clusters flatten under a CO environment, suggesting that the stabilization gained via the adsorption energy of CO serves as a thermodynamic driving force to offset the stability loss by overwetting of the cluster to the support.” (pg. 8)

Comment 3

Looking at Figure 4a-c and Figure 4d a big shift in the CO stretching frequency is observed. This must be due to a change of adsorption site due to coverage effects. Something is said, but not clearly stated and not in the right place.

Response 3

This an excellent point. The spectra shown in Figure 4d is dominated by CO adsorbed on lower frequency bridge and threefold sites, whereas Figure 4a-c are predominantly dominated by higher frequency atop sites. The Monte Carlo simulations indicate that CO molecules tend to

preferentially adsorb on the bridge and threefold sites on larger clusters versus smaller clusters. This phenomenon is also observed in the cited literature. We comment on this by adding the following statement:

“It is interesting that the dominant peak in the spectra of Error! Reference source not found.d (corresponding to Pd₂₀), centered in the ~1650 cm⁻¹ regime, is blue shifted when compared to the spectra in Error! Reference source not found.c (corresponding to Pd₁₀). This can be rationalized by CO preferentially adsorbing on lower wavenumber bridge and threefold sites on the Pd₂₀ cluster, while predominantly occupying higher wavenumber atop and bridge sites on Pd₁₀. The preferential adsorption on threefold and bridge sites on larger Pd clusters is also observed in the literature²⁵.” (pg. 14)

We explain the mentioned correlation in adsorption site type (atop, bridge, threefold) with the observed wavenumbers in the paragraph immediately following.

Comment 4

Figure 5. The spectra refer to supported clusters right? Please specify.

Response 4

Thank you for pointing this out. The spectra refer to supported clusters. This has been clarified in the figure caption as follows:

“Figure 1. Synthetic spectra of a system containing equal fractions of supported Pd monomers, dimers, and trimers saturated with CO.” (pg. 17)

Comment 5

Figure 6. The spectra refer to gas-phase clusters? Please specify.

Response 5

Thank you for pointing this out. The spectra refer to the supported clusters and is the same system as the one referred to in Figure 4. This has been clarified in the figure caption as follows:

“Figure 2. Synthetic spectra deconvolution of a system containing equifractions of supported Pd₁, Pd₂, and Pd₃ saturated with CO.” (pg. 22)

Comment 6

Page 15. The sentence “detailed experimental surface and nanocluster characterization is only achievable with ordered adsorbate overlayers on single crystals” (p. 15) is unclear or even

wrong. Several examples of characterization of supported nanoclusters exist for disordered systems and are based on various techniques, not only IR.

Response 6

Thank you for pointing this out. We have changed the sentence to reflect the relative difference in difficulty of experimental characterization of systems with ordered versus disordered overlayers. This change is reflected as follows:

“Detailed experimental surface and nanocluster characterization is difficult to achieve for working materials and is often limited to simpler ordered adsorbate overlayers on single crystals⁴⁰.” (pg. 23)

Comment 7

The comparison of the simulated IR spectra with the experimental ones is done for a perfect CeO₂ surface but experimentally it is difficult to prepare defect-free ceria. It is possible that the oxygen vacancies do not affect in a major way the IR spectra, but this depends on the nature of the support, stoichiometric or reduced. What is the level of chemical reduction of the samples used to obtain the IR spectra?

Response 7

This an excellent point. Our framework is applicable to both systems with and without defects and only improves with additional orthogonal characterization techniques to improve the accuracy of the modeled system. We assumed a defect free CeO₂(111) surface for illustration purposes. While we have not investigated the effect of oxygen vacancies on the effect of the IR spectra, this is an important future research direction.

The samples synthesized by Spezzati et al. used in our first deconvolution example were reduced at 300 °C in H₂, but due to limitations of the provided characterization, the defect fraction is unknown. The samples synthesized by Binet et al. used in our second deconvolution example were reduced at 423 K in H₂. The authors utilized both methanol adsorption and TCNE adsorption on clean CeO₂ without Pd and concluded that “treatment of CeO₂ (without Pd) by H₂ at 423 K does not lead to any observable reduction of the cerium surface sites”.

We have added the following to comment on the oxidation state of each sample, as well as the limitations of our existing defect-free CeO₂ dataset.

“We also note that we model a defect free CeO₂, while the extent of reduction of the support of the sample is unknown due to limited characterization. The effect of oxygen vacancies on IR spectra is undoubtedly an important topic of future research.” (pg. 24)

“We note that the sample was reduced at 423 K in H₂ but the authors were able to deduce, via methanol and TCNE adsorption, that there was no observable reduction of the support.” (pg. 26)

Comment 8

In summary, this paper reports interesting and novel results, but in my view it cannot be published in present form and needs to be rewritten to large extent. I appreciate the attempt to be succinct and concise, but this must not be at the expense of clarity.

Response 8

We appreciate the constructive feedback of the reviewer that helped us improve the manuscript.

Reviewer 3

Characterization of supported clusters and nanoparticles is one main challenge in heterogeneous catalysis research. The size and structure of small, supported clusters are generally unknown and difficult to assess, in particular, in the presence of reactants. IR spectroscopy is a common technique, which allows for in situ information although the interpretation of the data generally is difficult.

The manuscript describes a computational study where a technique is outlined where vibrational IR data from CO is used to obtain information on the structure of small Pd clusters supported on CeO₂. The discussed methodology is interesting and contains different steps where statistical techniques are combined with DFT calculations. The use of many structures for one cluster size is important, which considers that systems at elevated temperatures could be structurally excited. The work is suitable for Nature Communications once the Authors have addressed the following points, which mainly concern the performance of the method with respect to actual experimental data.

Response: We appreciate the positive comments of the referee.

Comment 1

In the Introduction, the Authors state that IR-spectroscopy is used to study single-atom catalysts. IR-spectroscopy is a traditional technique to characterize catalysts of many types of metal particles supported on oxides; not only single-atom catalysts.

Response 1

Thank you for pointing this out. We have revised the sentence to reflect that IR spectroscopy is used to study many types of oxide supported metals, and not only single-atom catalysts. This change has been reflected as follows:

“Excitations, probed via infrared (IR) spectroscopy⁸, are sensitive to interactions between adsorbates and metals, and have been extensively used to study supported metal oxides and single-atom catalysts⁹⁻¹¹.” (pg. 2)

Comment 2

It is stated that IR-spectroscopy “occasionally” can be used for in-operando studies. IR-spectroscopy is probably one of the mostly used techniques for in-operando studies.

Response 2

Thank you for pointing this out. We have revised the sentence to reflect that IR is one of the most used techniques, rather than an occasionally used technique, for in-operando studies. This change has been reflected as follows:

“They can accurately capture adsorbate normal vibrational modes, account for coverage effects, and can be used in-operando.” (pg. 2)

Comment 3

It is stated that assignments are done with single crystal surfaces. It is true for some cases, however, many types of well-defined system, i.e. inorganic complexes, are commonly used to aid the assignments.

Response 3

This is an excellent point. We have included examples of other well-defined systems, like inorganic complexes used in homogeneous catalysis, to aid in the assignment of infrared spectroscopic peaks, especially for SACs. We have added the following sentence to reflect this change:

“Inorganic complexes in the form of homogeneous catalysts have also served as molecular analogs to mononuclear metal active sites of SA catalysts to aid in peak identification¹²⁻¹⁴.” (pg. 2)

Comment 4

It is stated that the proposed method is “an important step in intelligent catalyst design”. The method is rather a step to aid catalyst characterization.

Response 4

Thank you for pointing this out. We have revised the sentence to show that our methodology is an important step in catalyst characterization. This change has been reflected as follows:

“Advances in addressing these challenges is imperative to improving catalyst characterization and eventually catalyst performance^{9,10}.” (pg. 2)

“Our results obtained directly from the deconvolution of IR spectra with little to no a priori assumptions are consistent with those made from other orthogonal characterization techniques. The methodology is an important tool in catalyst characterization towards closing the materials gap.” (pg. 3)

Comment 5

On line 143 it is stated: “We ignore vibrational contributions to the calculated enthalpies, as the vibrational entropy of adsorbed CO is less than 0.03 eV on metals”. Perhaps the Authors are mixing enthalpy and entropy, thus it is difficult to understand what they mean. It is not clear from text if zero-point corrections are included. The adsorbed CO molecules have many modes, so the zero-point corrections should be included.

Response 5

Thank for you pointing this out. We computed the Gibbs free energy of each CO-Pd/CeO₂ configuration. The entropic contributions to the free energy can be separated into configurational and vibrational contributions. The former is accounted for through Grand Canonical Monte Carlo sampling, while the latter can be computed from statistical mechanics and has been left out. The following text has been added in the Results to clarify this point:

“The entropic contributions to the free energies can be decomposed into the respective configurational and vibrational contributions. We ignored vibrational entropy contributions, as the vibrational entropy of adsorbed CO is less than 0.03 eV on metals^{23–26}. Configurational entropy is explicitly accounted for by the Metropolis sampling scheme.” (pg. 9)

While we agree that adsorbed CO molecules have many vibrational modes, we found that zero-point corrections were not needed as when comparing differences in free energies between identical cluster sizes. The computed CO frequencies and coverages were similar for identical cluster sizes, leading to cancellation of ZPEs. We note that at high temperatures, vibrations of Pd atoms become significant but we have left these out to reduce computational time. The following text has been added to the Methods to reflect this:

“Zero-point energy (ZPE) corrections to the electronic energies were not needed as adsorbate frequencies were similar for identical cluster sizes, thus leading to similar ZPE corrections that cancelled out when comparing free energy differences. We ignored the vibrational contributions of the Pd atoms to reduce computational time but note that these vibrations may be important at high temperatures⁴³.” (pg. 30)

Comment 6

It seems as the Authors compute energies, not enthalpies.

Response 6

Thank for you pointing this out. We have clarified that we are computing and comparing Gibbs free energies between different CO-Pd/CeO₂ configurations, and not enthalpies. This has been addressed in Response 5.

Comment 7

It seems as the entropy contributions from the adsorbed CO molecules on the clusters are neglected, which is questionable.

Response 7

Thank for you pointing this out. This has been addressed in Response 5.

Comment 8

It looks like Pd₅ can hold five CO molecules. It is written (CO)₄/Pd₅ below the figure.

Response 8

Thank for you pointing this out. The figure has been appropriately corrected to reflect (CO)₅/Pd₅. (pg. 11)

Comment 9

On line 233 it is stated that noise is added partly to account for errors in the computed wavenumbers. The error is generally systematic (which is used in the scaling), this the DFT error is not white noise.

Response 9

This is an excellent point. We agree that errors in DFT computed frequencies are indeed systematic and are usually underestimated due to the infinite metal mass approximation. We note that as we are sampling shifted frequencies through the linear scaling factor from a Gaussian prior, the difference of two Gaussian random variables is normally distributed as well, so it is not unreasonable to lump this within our Gaussian noise term. However, we concede there may be better mathematical ways to express this error in future work. We have adjusted the description of the noise added to reflect that it is entirely random in nature. The change is reflected as follows:

“The error term, ϵ , is entirely random in nature and is intended to account for (1) background noise absent from the computational spectra, (2) DFT error in computed frequencies, and (3) spectral intensities for clusters/adsorbates not accounted for in the low-energy ensemble. We

note that the DFT errors in computed frequencies, are usually systematically underestimated in nature due to the infinite mass approximation and may not be entirely represented in the proposed mathematical form.” (pg. 18)

Comment 10

In Figure 7, the Authors compare their method to experiments from the literature. However, the experiment seems to have been obtained on a sample that was cooled to 50 C in the presence of O₂ (Spezzatti et al). Is Pd purely metallic when exposed to CO?

Response 10

This is an excellent point. Spezzatti et al. showed that CO can reduce the PdO phase at 50 C. Peterson et al. also demonstrated that oxidized Pd atoms are the active sites for low-temperature CO oxidation; however, during the reaction, the oxidized Pd becomes reduced and leads to loss of activity. It is possible that the sample may contain some oxidized clusters. We note that the effect of metallic versus oxidized metal clusters on IR signatures is significant and this provides further evidence that the catalyst is mainly reduced. We have included the following statement to reflect the uncertainty of the oxidation state of Pd:

“We note that the oxidation state of Pd is uncertain and, as a result, the clusters may not be entirely metallic. However, there is significant evidence (by the authors and in literature) that small PdO clusters as well as single atoms can be reduced by CO at low temperatures¹⁸, so we assume that Pd is metallic in our deconvolution process. The comparison to the experimental spectra provides further evidence for this.” (pg. 24)

Comment 11

To use the computational spectral for deconvoluting the experimental spectra and obtain information about cluster sizes it is important that the computational data can reproduce well defined systems. The Authors have used Pd(100) and Pd(111) as references and scaled wavenumbers to reproduce the experimental data. It should be mentioned in the main text that this procedure is used. Wavenumbers computed in DFT are often wrong by some 50 cm⁻¹, so a scaling procedure is needed. However, which references that have been used is not clear in the manuscript or the SI, which affects whether a proper scaling has been applied.

The computed modes are compared to experiments in the Table S1. The experimental values are taken from references 10 and 11. The original data in Ref 10 is measured in under electrochemical conditions and not used in Table S1. The data that is used is from UHV work by Bradshaw only cited in Ref. 10. Taking the example with Pd(100). Bradshaw (Surface Science 1982) states a saturation coverage of 0.56 for the peak corresponding to 1964 cm⁻¹. In the manuscript the calculations appear to have been done at a coverage of 1. For a higher coverage

of 0.81, Bradshaw gives a wavenumber of 1997 cm^{-1} . This puts some doubts on the comparison to the experiments and the scaling. Because of the coupling of modes, the Authors should present the full computed spectra for the surface systems with wavenumbers and intensities.

Response 11

Thank for pointing this out. Frequency scaling factors are fitted during the spectra deconvolution procedure; however, to prevent overfitting, we utilize computed scaling factors (and the associated uncertainty) from adsorbates on well-defined single crystals as informative priors to essentially bound our prediction. This serves as a regularization technique and provides reasonable estimates for the extent of error in DFT computed frequencies. It was previously unclear in the main text, so we have added the following in the main text to showcase this:

“DFT-computed frequencies are often systematically underestimated, and as a result, it is customary to utilize linear scaling factors fit to experimental data to correct for these errors. Linear frequency scaling factors are used for our computed primary spectra, which are variable and optimized during the fitting procedure. Each cluster can be thought of as having a distribution of spectroscopic signatures stemming from the uncertainty of DFT, in which the best spectra is chosen during the fitting procedure. Scaling factors computed for adsorbates on well-defined single crystals are used as informative priors to regularize and prevent overfitting. These calculated factors serve as reasonable estimates for the extent of error in DFT frequencies. More information on the construction of linear scaling factors can be found in the Supporting Information.” (ppg. 13-14)

We adjusted the caption of Figure 4 to reflect that linear scaling factors have not been used in the plotted primary spectra.

We have also clarified this point in the SI accordingly as follows:

“The “true” linear scaling factor is fit during the Bayesian Inference procedure and utilizes the bounds for the scaling factor on well-defined Pd crystals as a regularization mechanism to prevent overfitting.” (ppg. S5-S6)

To demonstrate the efficacy of linear scaling factors, we have included an example of reproducing experimental spectra from DFT with and without the use of linear scaling factors, rather than showing wavenumbers alone in Figure S6. We have clarified that we replicate the exact overlayers and coverages in our DFT calculations. We have also included individual references for each cited frequency for better clarity. (ppg. S17-S18)

REVIEWERS' COMMENTS

Reviewer #2 (Remarks to the Author):

I read the answers and comments of the authors to the points raised in the original reports and I think the paper has been considerably improved and should be considered for publication.

Reviewer #3 (Remarks to the Author):

The Authors have considered comments and the manuscript is suitable for publications once the following points are addressed (keeping the numbering from the previous round.)

1) The revised sentence is still too limited. IR spectroscopy is commonly used to study, supported metal particles, supported metal oxide particles, metal oxides and eventually single-atom catalysts.

2) The second sentence in the second paragraph is uncommonly formulated. IR-spectroscopy probe normal vibrational modes. The Authors write that IR-spectroscopy “accurately capture” normal vibrational modes.

4) The term “orthogonal characterization techniques” on page 3 is strange. Different physical characterization techniques could hardly be regarded as orthogonal. The techniques probe in different ways the potential energy surface between atoms.

5) The Authors state that the vibrational entropy of adsorbed CO is less than 0.03 eV on metals and give 4 references (23-26). I had a look in Reference 26 and do not find any discussion on the vibrational entropy of adsorbed CO. Are the references correct?

Are the Authors only considering the CO stretch vibration? The vibrational entropy is about 0.1 eV at room temperature for CO adsorbed on Pt(111) using the harmonic approximation, which clearly underestimates the entropy of adsorbed CO.

7) Again, I do not understand using the configurational entropy but not the vibrational entropy. How large is the configurational entropy for, for example, (CO)₇/Pd₇?

11) It is not clear what Figure S6 shows. It is stated that it is the relative IR intensities, and the experiments are probably taken from Reference 11 (as in Table S1). (There should be a reference also in Figure S6 regarding the origin of the experiments.) It seems as if the Authors simply have broadened the wavenumbers taken from the reference 11, thus it is not IR intensities. Which coverage are the Authors considering when doing the comparisons the experiments?

Summary of Overall Changes in Response to Reviewer Comments

- We improve the overall explanations regarding comparing experimental/computational infrared spectra, and entropic contributions to free energy differences to improve the technical clarity of the paper.
- We enhance the overall quality and clarity of figures and captions following the appropriate artwork guidelines.

Point-by-Point Comments and Responses to Reviewers' Comments

Reviewer 2

I read the answers and comments of the authors to the points raised in the original reports and I think the paper has been considerably improved and should be considered for publication.

Response: We appreciate the positive appraisal of the referee.

Reviewer 3

The Authors have considered comments and the manuscript is suitable for publications once the following points are addressed (keeping the numbering from the previous round.)

Response: We appreciate the positive comments of the referee.

Comment 1

The revised sentence is still too limited. IR spectroscopy is commonly used to study, supported metal particles, supported metal oxide particles, metal oxides and eventually single-atom catalysts.

Response 1

Thank you for pointing this out. We have revised the sentence to reflect that IR spectroscopy is used to study many types of materials, such as supported metal particles, supported metal oxides, and metal oxides, in addition to single-atom catalysts. This change has been reflected as follows:

“Excitations, probed via infrared (IR) spectroscopy¹¹, are sensitive to interactions between adsorbates and metals, and have been extensively used to study the structure of metal oxides, supported metal particles and metal oxides, as well as single-atom catalysts¹²⁻¹⁴.” (pg. 2)

Comment 2

The second sentence in the second paragraph is uncommonly formulated. IR-spectroscopy probe normal vibrational modes. The Authors write that IR-spectroscopy “accurately capture” normal vibrational modes.

Response 2

Thank you for pointing this out. We have revised the wording of the sentence to reflect that IR-probes normal vibrational modes, rather than capture. This change has been reflected as follows:

“They can accurately probe adsorbate normal vibrational modes, account for coverage effects, and can be used in-operando.” (pg. 2)

Comment 3

The term “orthogonal characterization techniques” on page 3 is strange. Different physical characterization techniques could hardly be regarded as orthogonal. The techniques probe in different ways the potential energy surface between atoms.

Response 3

Thank for you pointing this out. We have replaced the term “orthogonal characterization techniques” simply with “characterization techniques”. This change is reflected as follows:

“Our results obtained directly from the deconvolution of IR spectra with little to no a priori assumptions are consistent with those made from other characterization techniques. The methodology is an important tool in catalyst characterization towards closing the materials gap.” (pg. 3)

Comment 4

The Authors state that the vibrational entropy of adsorbed CO is less than 0.03 eV on metals and give 4 references (23-26). I had a look in Reference 26 and do not find any discussion on the vibrational entropy of adsorbed CO. Are the references correct?

Are the Authors only considering the CO stretch vibration? The vibrational entropy is about 0.1 eV at room temperature for CO adsorbed on Pt(111) using the harmonic approximation, which clearly underestimates the entropy of adsorbed CO.

Response 4

Thank you for pointing this out. We have clarified that the difference in vibrational entropies between different CO adsorption site types is in the order of 0.03 eV on metals. This number is estimated in accordance with Reference 29, which finds that the largest differences in vibrational entropy of adsorbed CO on different site types is approximately 0.085 meV/K, which translates to <0.03 eV at the temperature of 323 K. As we are analyzing free energy differences (rather than absolute energies), the vibrational entropic contribution to the free energies is ignored. This change is reflected as follows:

“We ignored vibrational entropy contributions to the free energy differences, as the change in vibrational entropy of adsorbed CO on different sites is typically less than 0.03 eV at 323 K on metals^{29–32}.” (pg. 9)

Comment 5

Again, I do not understand using the configurational entropy but not the vibrational entropy. How large is the configurational entropy for, for example, (CO)₇/Pd₇?

Response 5

Thank you for pointing this out. As mentioned in the previous response, the contribution of vibrational entropy to the free energy difference is negligible at the temperatures analyzed in this study. Configurational entropy is explicitly accounted for in the Metropolis sampling scheme as it explores the combinatorial space of CO/Pd. The distributions of the free energies shown in Figure 2b, for example, accounts the numbers of relative number of configurations that have a particular free energy. Thus, it is not needed to explicitly calculate configurational entropy from statistical thermodynamics, where $S = -k_B \sum_{n=1}^W P_n \ln(P_n)$, k_B is the Boltzmann constant and P_n is the probability of a given state n .

Comment 6

It is not clear what Figure S6 shows. It is stated that it is the relative IR intensities, and the experiments are probably taken from Reference 11 (as in Table S1). (There should be a reference also in Figure S6 regarding the origin of the experiments.) It seems as if the Authors simply have broadened the wavenumbers taken from the reference 11, thus it is not IR intensities. Which coverage are the Authors considering when doing the comparisons the experiments?

Response 6

Thank for pointing this out. Figure S6 reproduces the spectra shown in Reference S11 (which corresponds to the frequencies and coverages shown in Table S1). Intensities are obtained from digitization of the spectra. The purpose of Figure S6 is to show that frequency scaling factors allow computational spectra to better match experimental spectra.